# VeriThinker:
# Learning to Verify Makes Reasoning Model Efficient

**Zigeng Chen, Xinyin Ma, Gongfan Fang, Ruonan Yu, Xinchao Wang**[*]
National University of Singapore
zigeng99@u.nus.edu, xinchao@nus.edu.sg

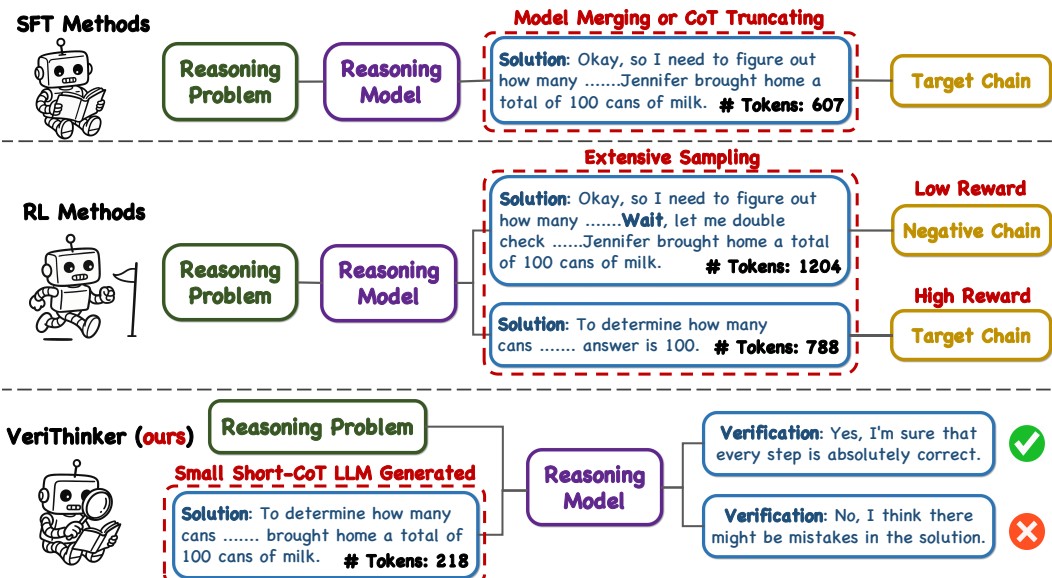

Figure 1: The key distinction between VeriThinker and traditional SFT or RL-based long-to-short methods. We uniquely train LRMs on an auxiliary CoT verification task, achieving effective CoT compression without relying on synthetic target reasoning chains.

## Abstract

*Large Reasoning Models* (LRMs) excel at complex tasks using *Chain-of-Thought* (CoT) reasoning. However, their tendency to overthinking leads to unnecessarily lengthy reasoning chains, dramatically increasing inference costs. To mitigate this issue, we introduce VeriThinker, a novel approach for CoT compression. Unlike conventional methods that fine-tune LRMs directly on the original reasoning task using synthetic concise CoT data, we innovatively fine-tune the model solely through an auxiliary verification task. By training LRMs to accurately verify the correctness of CoT solutions, the LRMs inherently become more discerning about the necessity of subsequent self-reflection steps, thereby effectively suppressing overthinking. Extensive experiments validate that VeriThinker substantially reduces reasoning chain lengths while maintaining or even slightly improving accuracy. When applied to DeepSeek-R1-Distill-Qwen-7B, our approach reduces reasoning tokens on MATH500 from **3790** to **2125** while improving accuracy by 0.8% (**94.0%** to **94.8%**), and on AIME25, tokens decrease from **14321** to **10287** with a 2.1%

---

[*]Correspoding Author

39th Conference on Neural Information Processing Systems (NeurIPS 2025).

accuracy gain (**38.7%** to **40.8%**). Additionally, our experiments demonstrate that VeriThinker can also be zero-shot generalized to speculative reasoning. Code is available at `https://github.com/czg1225/VeriThinker`

# 1 Introduction

*Large Reasoning Models* (LRMs), such as GPT-o1 [20], DeepSeek-R1 [13], Kimi-k1.5 [54], and QwQ [55], have shown exceptional performance in tackling complex reasoning tasks. Their success stems from the ability to generate effective *Chain-of-Thought* (CoT) reasoning [59, 32]. By decomposing intricate problems into manageable steps and meticulously verifying each intermediate result, these models achieve superior reasoning performance. Furthermore, this test-time scaling method has demonstrated notable potential beyond reasoning tasks, extending its impact to diverse fields such as vision-language models [19, 63, 72, 28], image generation [40, 21, 62], and video synthesis [7, 33].

However, the long reasoning chains generated by LRMs often contain numerous redundant self-checking steps—a phenomenon commonly known as "overthinking" [5]. Specifically, these models incorporate frequent self-verification steps to ensure the accuracy of their reasoning, but most of these self-verifications prove ineffective or even add confusion to the reasoning process. This extensive overthinking dramatically inflates inference costs and hinders the efficient deployment of LRMs.

Several recent studies [41, 14, 60, 35, 43, 38, 71, 2, 1] have focused on compressing lengthy reasoning chains into shorter, more concise forms. These methods typically generate concise reasoning chain data through extensive sampling, model merging, or selective truncation of original long reasoning chains. Subsequently, these synthetic reasoning chains serve as targets, with *Supervised Fine-Tuning* (SFT) [58] or *Reinforcement Learning* (RL) used to align the LRM's output distribution closely to these target chains. Thus, the performance of these compression strategies heavily depends on the quality and quantity of the synthesized reasoning chain data. However, generating high-quality concise reasoning chains is computationally intensive and time-consuming. Moreover, simultaneously achieving brevity while preserving essential self-reflection steps remains challenging. Consequently, models compressed through these methods struggle to balance conciseness and high accuracy, especially when tackling highly complex reasoning tasks. A key question arises: *Can we effectively compress lengthy reasoning chains without relying on synthetic target reasoning chains?*

**Our Approach.** To address this challenge, we introduce VeriThinker, a straightforward yet effective method for CoT compression without the dependence on synthetic target data. VeriThinker's core innovation lies in a novel fine-tuning strategy named *Supervised Verification Fine-Tuning* (SVFT). Unlike previous methods that directly fine-tune LRMs on reasoning tasks through SFT or RL, SVFT uniquely fine-tunes the model on an auxiliary verification task to achieve CoT compression. Specifically, we create a CoT verification dataset, providing the LRM with question-solution pairs and training it to verify the correctness of the CoT solution through binary classification. By learning to distinguish between correct and incorrect solutions, the LRM can more accurately determine when self-reflection is necessary. Empirical studies demonstrate that after SVFT, the model significantly reduces unnecessary double-checking of correct reasoning steps while slightly increasing verification of incorrect steps. This effectively mitigates overthinking and maintains or slightly improves reasoning accuracy. Furthermore, by utilizing its ability to recognize correctness, VeriThinker can also generalize to solution-wise speculative reasoning for higher inference throughput.

We evaluate VeriThinker on three state-of-the-art reasoning models: DeepSeek-R1-Distill-Qwen-7B, DeepSeek-R1-Distill-Qwen-14B, and DeepSeek-R1-Distill-Llama-8B [13]. Extensive experiments show that fine-tuning solely on the CoT verification task enables LRMs to substantially reduce token usage while preserving high accuracy, even on extremely challenging problems like the AIME dataset. Additional experiments show that VeriThinker can also be applied to speculative reasoning, achieving a significant throughput increase when using a short-CoT LLM as the draft model.

In conclusion, we present VeriThinker, a simple yet effective CoT compression method that eliminates the need for synthetic target chain data. Key to our approach is the proposed SVFT, which innovatively mitigates the overthinking problem by fine-tuning the LRM exclusively on an auxiliary CoT verification task. Comprehensive experiments demonstrate that VeriThinker efficiently shortens reasoning chains while preserving or slightly improving model accuracy, even on highly challenging reasoning tasks. Additionally, our method can also generalize to speculative decoding, achieving significant improvements in inference throughput.

## 2 Related Works

**Chain-of-thought.** Chain-of-thought [59, 32] enables large reasoning models (LRMs) to solve complex reasoning tasks, such as mathematical and coding problems, by leveraging inference-time scaling laws. Advanced LRMs, including GPT-o1 [20], DeepSeek-R1 [13], Kimi-k1.5 [54], and QwQ [55], have set new benchmarks in reasoning capabilities. These models utilize reinforcement learning to encourage multiple self-reflection steps during inference, substantially increasing response accuracy. Furthermore, other methods, such as self-consistency [57], beam search, and Monte Carlo Tree Search (MCTS) [22, 12], have been employed to facilitate parallel inference scaling laws, further enhancing LRM reasoning effectiveness. Beyond language models, the principles underlying CoT have also demonstrated significant performance improvements in domains like visual reasoning [19, 63, 72], image generation [40, 21, 62], and video synthesis [7, 33].

**Efficient Reasoning Models.** Despite these substantial gains, the tendency of LRMs towards overthinking often results in excessively extended reasoning chains, significantly raising inference costs. Recent research has investigated methods for compressing CoT to enable more efficient reasoning processes. Specifically, [5] comprehensively analyzes the overthinking phenomenon and employs SimPO [42] to fine-tune LRMs. Kimi-K1.5 [54] proposes weight merging strategies between long-CoT and short-CoT models and introduces length-penalized loss functions to effectively compress CoT. Additional studies [41, 14] achieve lengthy-controllable CoT compression, while others [60, 35, 43, 69] synthesize concise reasoning chain data to guide long-to-short compression via SFT. Methods [38, 71, 2, 1, 26, 17] adopt reinforcement learning and extensive sampling to reduce the number of tokens required in CoT processes. Furthermore, adaptive methods [68, 37, 50, 10] have been proposed to conditionally activate long-CoT modes based on input query. Simpler yet effective prompt-guidance methods for direct CoT compression have also been explored [36, 65, 24, 53]. Additionally, there is increasing interest in converting explicit CoT processes into latent CoT [9, 8, 49, 11, 15]. However, these latent approaches currently face significant challenges about accuracy degradation.

Beyond CoT compression, other techniques involving model compression and optimized decoding have also gained prominence. Knowledge distillation approaches have allowed smaller reasoning models to approximate the performance of larger counterparts effectively [64, 4, 27, 74]. Additionally, [51, 34, 73] explore model pruning and quantization techniques on LRMs. Decoding optimization techniques have also been widely adopted to accelerate inference [56, 52, 39, 64, 67]. Methods such as speculative decoding [31, 46] and parallel inference strategies [45, 44] demonstrate a notable decoding acceleration ratio, further enhancing the practical applicability of LRMs.

## 3 Methods

### 3.1 Problem Setup

Recent efforts in improving the efficiency of LRMs have focused on compressing lengthy reasoning chains into shorter, more concise reasoning sequences without redundant over-thinkings. Formally, let $M_\theta$ represent an LRM parameterized by $\theta$. Given a query $q$, the model generates a long reasoning chain $C_l$, obtained by sampling from the model's conditional output distribution: $C_l \sim M_\theta(\cdot \mid q)$. We hypothesize the existence of an ideal concise reasoning chain $C_i$, characterized by minimality in length and maximal retention of essential self-verification steps, ensuring the correctness of the final solution. The objective of CoT compression is thus to finetune $M_\theta$ such that the generated reasoning chain closely approximates the ideal concise chain $C_i$:

$$\min_\theta \mathbb{E}_q[D(M_\theta(\cdot \mid q), C_i)], \qquad (1)$$

where $D$ denotes a suitable distance metric in the distribution space.

A commonly employed approach involves constructing a synthetic short reasoning chain $C_s$. In SFT-based variants [60, 35, 43, 69, 41], this is typically done via model merging or by selectively truncating the original long chain data, whereas RL-based approaches [38, 71, 2, 1, 26, 17] rely on extensive sampling. Under the assumption that the synthetic chain $C_s$ is distributionally close to the ideal chain $C_i$, they train the LRMs either by supervised fine-tuning on $C_s$ or by reinforcement learning—rewarding the shorter sequences $C_s$ more highly than the longer ones $C_l$.

In essence, these approaches regard the synthesized short CoT sequences as the target reasoning chain and train the model to approximate them. Therefore, their efficacy depends critically on both the quality and volume of the generated $C_s$. However, generating high-quality short chains is extremely costly in computing and time, which limits efficient scalability. Furthermore, synthetic short reasoning chains typically struggle to simultaneously achieve brevity and retain the essential self-reflection steps, resulting in a mismatch from the ideal distribution $C_i$. This discrepancy often inevitably leads to substantial degradation in the model's reasoning capabilities during the long-to-short process, particularly when confronted with complex, high-difficulty problems.

A critical problem arises: Can we devise a method that effectively compresses an LRM's overextended reasoning chains while preserving its original reasoning ability, without relying on explicitly synthesized target chains?

## 3.2   Supervised Verification Finetuning

To address the aforementioned issue, we begin by revisiting the fundamental principles underlying LRM's overthinking. During reasoning, when determining whether to engage in additional self-reflection, the LRM essentially operates as a binary classifier that evaluates the correctness of previously generated reasoning steps. Formally, the LRM encodes a prior solution into a hidden state, denoted as $h$, and the language modeling head (LM head) subsequently classifies this state as either *correct* or *incorrect*. If the classification is *correct*, the model proceeds to subsequent reasoning steps; if *incorrect*, it triggers further self-verification.

The prevalence of overthinking stems from insufficient accuracy in this binary classification task. Specifically, the hidden state $h$ fails to adequately encode the critical information required for precise correctness judgments. Consequently, LRMs frequently misclassify correct solutions as incorrect, leading to unnecessary self-reflection steps. Let $p(\text{acc} \mid h)$ denote the probability that the LRM accurately classifies the correctness of a prior solution. By increasing $p(\text{acc} \mid h)$, redundant reflections are progressively reduced, preserving only essential verification steps for genuinely incorrect solutions. In the ideal case where $p(\text{acc} \mid h) = 100\%$, the LRM outputs the optimal reasoning chain $C_i$, invoking self-reflection solely when prior solutions are indeed erroneous.

Existing methods that employ synthetic target chains for CoT compression implicitly aim to maximize $p(\text{acc} \mid h)$. These methods synthesize a concise target chain approximating the optimal scenario where $p(\text{acc} \mid h) \approx 100\%$. As the LRM's output distribution aligns with this target, $p(\text{acc} \mid h)$ naturally increases, thereby mitigating overthinking. However, such approaches are inefficient and indirect, as most tokens in the synthetic chains contribute negligibly to improving $p(\text{acc} \mid h)$. Consequently, these methods yield only marginal gains in $p(\text{acc} \mid h)$. Moreover, the probability of triggering self-reflection decreases even when errors occur in earlier steps, inadvertently leading to performance degradation on highly complex problems.

Since the primary cause of overthinking in LRMs arises from their inability to accurately determine the correctness of a solution, a natural and intuitive strategy is to explicitly train LRMs to discern solution correctness, thus fundamentally addressing the overthinking issue. Motivated by this insight, we introduce VeriThinker, a straightforward yet effective method for CoT compression. The key idea of VeriThinker is a novel fine-tuning strategy, termed *Supervised Verification Fine-Tuning* (SVFT). Distinct from prior approaches, SVFT does not rely on optimizing the model to match a target reasoning chain distribution. Instead, it aims to directly enhance the model's capability to classify solutions as correct or incorrect, optimizing the binary classification accuracy $p(\text{acc}|h)$. Specifically, we fine-tune the reasoning model on an auxiliary CoT verification task, providing the LRM with a question paired with a CoT solution, and the model is explicitly trained to accurately classify whether this provided solution is entirely correct or not.

Firstly, we need to construct a CoT verification dataset, denoted as: $\mathcal{D}_{\text{verif}} = (q_i, s_i, v_i)$ where each data instance consists of a problem description $q_i$, a CoT solution $s_i$ without self-reflection steps, and a verification result $v_i$ indicating the correctness of the solution. The prompt for each instance is formed by the pair $(q_i, s_i)$, and the corresponding verification response $v_i$ is explicitly defined as:

$$v_i = \begin{cases} \text{"Yes, I'm sure that every step is absolutely correct.",} & \text{if } s_i \text{ is correct,} \\ \text{"No, I think there might be some mistakes in the proposed solution.",} & \text{otherwise.} \end{cases} \quad (2)$$

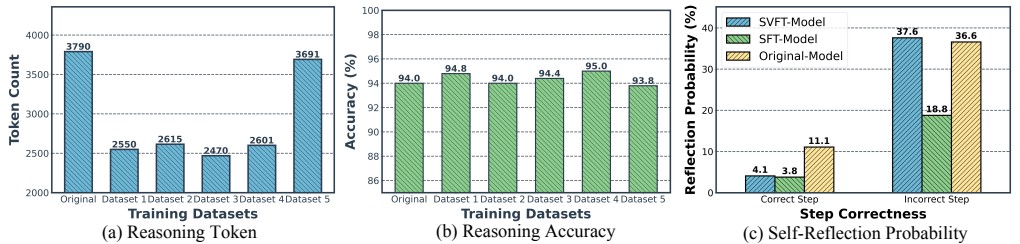

Figure 2: (a)-(b): Token counts and accuracy in reasoning tasks across different training datasets. (c) Probability of self-reflection during reasoning for correct and incorrect solutions.

Then we used this CoT-verification dataset to fine-tune the LRMs. During finetuning, only tokens within the verification response $v_i$ contribute to the training loss. This design effectively mitigates potential undesirable biases introduced by the input solution from affecting the model's output distribution. Formally, given a tokenized prompt-response pair $(x, y)$, the SVFT training loss is defined as:

$$\mathcal{L}_{\text{SVFT}} = -\sum_{t \in y} \log p_\theta(y_t | x, y_{<t})$$ (3)

where $p_\theta$ denotes the probability distribution output by the LRM parameterized by $\theta$.

Since the judgments of correct and incorrect in the data are represented as two completely fixed responses, and we only compute the loss on the response tokens, the entire SVFT essentially performs a binary classification on the question-solution pairs which is fundamentally similar to how the LRM decides whether to perform self-reflection after each step during reasoning.

Experiments demonstrate that by learning to verify the correctness of CoT solutions, SVFT effectively mitigates LRM's overthinking phenomenon. While significantly reducing the number of tokens required for reasoning, SVFT does not lead to noticeable degradation in reasoning capability.

### 3.3 Empirical Analysis

To better understand why SVFT facilitates effective compression of CoT without compromising the original reasoning capability of LRMs, we conducted additional experiments to thoroughly investigate the underlying mechanisms of SVFT.

**What Capabilities Does SVFT Impart to the Model?** A primary question we sought to explore was the specific competencies instilled in the LRM through SVFT. To this end, we constructed five distinct CoT-verification datasets and conducted SVFT on the R1-Distill-Qwen-7B model, observing differences in outcomes across these datasets. These five datasets shared identical prompts, each consisting of a question, a CoT solution, and concise instructions. However, we varied the response tokens in five distinct ways: **Dataset (1)**: Maintained the original setting. **Dataset (2)**: Reversed responses, assigning affirmative responses to incorrect solutions and negative responses to correct solutions. **Dataset (3)**: Simplified responses to single tokens, "Yes" and "No,". **Dataset (4)**: Further simplified responses to semantically neutral tokens "North" and "South." **Dataset (5)**: Randomly assigned responses irrespective of solution correctness.

Figure 4 (a) and (b) illustrate the changes in average tokens per response and accuracy on the MATH500 [16] dataset after SVFT training with the aforementioned datasets. We observed that, apart from Dataset (5), all other datasets showed similar trends: a significant reduction in reasoning chain length, accompanied by maintained or slightly improved accuracy. These findings suggest that SVFT is fundamentally akin to contrastive learning. The model learns the capability to distinguish differences between correct and incorrect solutions through binary classification rather than learning explicit correctness semantics. In contrast, Dataset (5), which utilized responses entirely unrelated to solution correctness, did not facilitate CoT compression. This indicates that SVFT does not merely cause the LRM to adopt a more concise expression style or to directly output conclusions, reinforcing our earlier conclusions.

**Conclusion:** In summary, SVFT empowers the LRM to differentiate between correct and incorrect solutions rather than intrinsically learning which solution is right. The core mechanism of SVFT closely resembles that of contrastive learning.

**Why Does SVFT Enhance CoT Compression?** Next, we investigate the reasons behind SVFT's effectiveness in achieving CoT compression. As SVFT trains the model to perform binary classification based on the correctness of CoT solutions, the model inherently learns to focus on the crucial elements in the CoT affecting solution correctness. We hypothesize that this enhanced attention allows the model, during reasoning tasks, to more accurately determine whether a self-reflection step is necessary, thereby eliminating redundant double-checking and enabling effective CoT compression.

To empirically validate this hypothesis, we randomly selected 50 problems from the MATH500 dataset and generated correct long CoT solutions for each problem using the R1-distill-Qwen-7B model. We extracted sub-solutions from each solution, spanning from the start until the appearance of the first pivot word "Wait." We then performed forward computations using the original R1-Distill-Qwen-7B, the SFT model and the SVFT model on these question-sub-solution pairs, examining the generation probabilities of the pivot word "Wait." As shown in Figure 4 (c), we found that both the SFT and SVFT models exhibit a significantly lower probability of generating the pivot word for correct sub-solutions compared to the original model.

Next, we manually introduced mistakes into each correct sub-solution—for instance, changing "$3 \times 7 = 21$" to "$3 \times 7 = 27$"—and evaluated the models' probabilities for generating the pivot word "Wait" again. In this case, the SVFT model showed no probability reduction but instead a marginal (1%) increase compared to the original model, while the SFT model continued to exhibit significantly reduced probabilities.

These results strongly support our hypothesis: SVFT enhances the LRM's precision in evaluating reflection necessity. Specifically, for the SVFT model, we observe: (1) substantial reduction in redundant verification when preceding steps are correct, and (2) slight increase in reflection probability when mistakes are introduced. This adaptive behavior enables effective CoT compression while preserving and occasionally improving reasoning accuracy. In contrast, the SFT model uniformly reduces reflection probability regardless of solution correctness, explaining its chain-shortening capability at the cost of accuracy degradation.

To further validate this hypothesis, we compared the changes in response token counts of the SVFT-finetuned model on the MATH500 dataset. For questions correctly answered by the SVFT model, the token count in the reasoning chain decreased by 41% compared to the original model. In contrast, for incorrectly answered questions, the token count decreased by only 8%. This experimental result further robustly supports our hypothesis.

**Conclusion:** In summary, SVFT achieves effective CoT compression primarily by enhancing the accuracy of LRMs' self-reflection decisions during reasoning. It minimizes unnecessary self-reflections for correct steps while retaining critical verification steps for mistaken steps, significantly reducing token usage without compromising reasoning accuracy.

### 3.4 Solution-wise Speculative Reasoning

As LRMs acquire the capability to verify solution correctness through SVFT, we leverage this advancement to propose a collaborative inference pipeline termed *Solution-wise Speculative Reasoning* (SSR). This approach significantly boosts inference throughput by orchestrating interactions between a short-CoT LLM and an SVFT-enhanced LRM.

Distinct from conventional token-wise speculative decoding methods [25, 3, 30, 61], where a smaller draft model proposes multiple candidate tokens subsequently verified in parallel by a larger LLM through token probability evaluation (accepting or rejecting each individual draft token), our approach introduces a novel solution-level paradigm specifically tailored for reasoning tasks. Given a problem, the short-CoT LLM first rapidly generates a concise solution candidate. This candidate is then assessed by the SVFT-enhanced LRM for correctness verification. If the proposed solution is deemed correct, it is immediately output as final answer; otherwise, the system activates the LRM's full long-CoT reasoning process to ensure accurate results.

This pipeline adaptively engages the LRM: For simple problems, the LRM only validates the draft solution without invoking costly long-CoT; For challenging problems, the LRM intervenes to ensure accuracy when the draft fails. By selectively triggering long-CoT reasoning based on problem difficulty and draft quality, our speculative reasoning pipeline achieves substantial throughput gains while preserving high reasoning accuracy.

Table 1: CoT compression performance. We evaluate the proposed VeriThinker on three advanced reasoning models, comparing token efficiency and accuracy against the original model and baseline methods over three mathematical reasoning benchmarks.

| Method | MATH500 | | AIME 2024 | | AIME 2025 | | Average | |
|--------|---------|---------|-----------|---------|-----------|---------|---------|---------|
| | Tokens ↓ | Accuracy↑ | Tokens ↓ | Accuracy↑ | Tokens ↓ | Accuracy↑ | Tokens ↓ | Accuracy↑ |
| *Deepseek-R1-Distill-Qwen-7B* | | | | | | | | |
| Original Model | 3791 | 94.0% | 13108 | 54.1% | 14321 | 38.7% | 10407 | 62.3% |
| Truncating | 2306 | 80.4% | 9550 | 45.3% | 10232 | 36.3% | 7363(-29%) | 54.0%(-8.3%) |
| Fast-Prompting | 2425 | 83.8% | 11867 | 53.3% | 13378 | 38.3% | 9223(-30%) | 58.5%(-3.8%) |
| CoT-Valve | 2440 | 92.4% | 11238 | 39.2% | 10884 | 30.4% | 8187(-22%) | 54.0%(-8.3%) |
| SFT | 2064 | 91.2% | 8843 | 49.1% | 9686 | 32.1% | 6864(-34%) | 57.5%(-4.8%) |
| **VeriThinker** | 2125 | **94.8%** | 9381 | **56.5%** | 10287 | **40.8%** | 7264(-30%) | **64.0%**(+1.7%) |
| *Deepseek-R1-Distill-Qwen-14B* | | | | | | | | |
| Original Model | 3529 | 95.2% | 11724 | 69.0% | 13409 | 49.6% | 9554 | 71.2% |
| Truncating | 2479 | 85.8% | 8658 | 57.1% | 9247 | 42.9% | 6795(-30%) | 61.9%(-9.3%) |
| Fast-Prompting | 2405 | 82.0% | 8288 | 64.0% | 10645 | 49.3% | 7113(-26%) | 65.1%(-6.1%) |
| CoT-Valve | 2102 | 91.0% | 8691 | 41.7% | 9558 | 25.0% | 6783(-29%) | 52.6%(-18%) |
| SFT | 2226 | 92.8% | 9021 | 58.3% | 10644 | 43.3% | 7297(-24%) | 64.8%(-7.4%) |
| **VeriThinker** | 2255 | **95.0%** | 7423 | **73.0%** | 9304 | **54.8%** | 6327(-34%) | **74.3%**(+3.1%) |
| *Deepseek-R1-Distill-Llama-8B* | | | | | | | | |
| Original Model | 4361 | 90.6% | 14005 | 44.6% | 14420 | 30.1% | 10928 | 55.1% |
| Truncating | 3193 | 81.6% | 11577 | 39.1% | 11635 | 28.6% | 8802(-20%) | 49.8%(-5.3%) |
| Fast-Prompting | 3378 | 86.2% | 12615 | 40.4% | 13292 | 29.4% | 9762(-11%) | 52.0%(-3.1%) |
| CoT-Valve | 4614 | 81.4% | 12650 | 23.3% | 12520 | 17.5% | 9928(-9%) | 40.7%(-14%) |
| SFT | 2439 | 81.2% | 9500 | 31.3% | 10136 | 26.0% | 7358(-33%) | 46.2%(-8.7%) |
| **VeriThinker** | 2953 | **89.9%** | 11285 | **46.9%** | 10557 | **29.7%** | 8265(-24%) | **55.5%**(+0.4%) |

# 4 Experiments

## 4.1 Experimental Setup

**Models.** To comprehensively evaluate our approach, we apply the proposed VeriThinker to three state-of-the-art long-CoT reasoning models with varying architectures and sizes, including DeepSeek-R1-Distill-Qwen-7B, DeepSeek-R1-Distill-Qwen-14B, and DeepSeek-R1-Distill-LLaMA-8B [13]. Additionally, we also evaluate our approach on three short-CoT models: Qwen-2.5-Math-7B-Instruct, Qwen-2.5-Math-1.5B-Instruct [66], and Qwen-2.5-7B-Instruct [66].

**Training and Evaluation Details.** For the proposed SVFT, we adopt *Low-Rank Adaptation* (LoRA) [18] for efficient fine-tuning, which significantly improves training efficiency and effectively mitigates catastrophic forgetting, as training and inference occur on different tasks. All training procedures are conducted using Hugging Face's SFTTrainer integrated with DeepSpeed ZeRO-2 optimization [47], distributed across four RTX 6000 Ada GPUs. For evaluation, inference results and throughput metrics are also obtained using RTX 6000 Ada GPUs with the vLLM inference framework [23]. We adhere to the models' default sampling configurations, specifically setting temperature to $0.6$ and top-$p$ to $0.95$. During inference, the prompt appended to each question is: *"Please reason step by step, and put your final answer with* `\boxed`*."* To evaluate LRM's performance, we report the reasoning accuracy and the average token count of the reasoning chain (between `<think>` and `</think>`).

**Datasets.** During the fine-tuning phase, we utilize our self-constructed CoT-verification dataset comprising approximately 340k question-CoT pairs, each labeled with correctness indicators. We provide more details about training set construction in the Appendix. For evaluation, we employ multiple mathematical benchmark datasets, including MATH500 [16], GSM8K [6], and two highly challenging competition datasets, AIME2024 and AIME2025. The results reported for each dataset represent averages computed over 2 to 16 independent runs, depending on the dataset's size.

**Baselines.** To demonstrate the superiority of our innovative method, we compare it against several conventional CoT compression approaches, including: (1) **Truncation**, which reduces the maximum allowable token length; (2) **Fast-Prompting**, a method leveraging prompt engineering to enforce reasoning completion within a specified token limit; (3) **CoT-Valve** [41], enabling a model to dynamically adjust the length of reasoning chains; and (4) **SFT** [58], a supervised fine-tuning method that uses synthesized concise CoT chains as targets for long-to-short compression. To synthesize concise reasoning chains for the SFT baseline, we follow CoT-Valve [1] by merging the base and

Table 2: CoT correctness verification results. We apply VeriThinker to three advanced LRMs and assess their verification accuracy on CoT solutions generated by two short-CoT models. For a comprehensive analysis, we report classification accuracy, precision, recall, and F1 score.

| Method | MATH500 | | | | GSM8K | | | |
|---|---|---|---|---|---|---|---|---|
| | Acc. ↑ | Precision ↑ | Recall ↑ | F1 Score ↑ | Acc. ↑ | Precision ↑ | Recall ↑ | F1 Score ↑ |
| *QWEN-2.5-MATH-1.5B-Instruct* | | | | | | | | |
| R1-Distill-Qwen-7B+VeriThinker | 88.4% | 0.926 | 0.924 | 0.926 | 93.6% | 0.964 | 0.962 | 0.963 |
| R1-Distill-Qwen-14B+VeriThinker | 90.6% | 0.953 | 0.927 | 0.940 | 95.7% | 0.971 | 0.978 | 0.975 |
| R1-Distill-Llama-8B+VeriThinker | 88.0% | 0.954 | 0.892 | 0.922 | 91.5% | 0.954 | 0.947 | 0.950 |
| *QWEN-2.5-MATH-7B-Instruct* | | | | | | | | |
| R1-Distill-Qwen-7B+VeriThinker | 89.4% | 0.946 | 0.931 | 0.939 | 94.8% | 0.981 | 0.965 | 0.973 |
| R1-Distill-Qwen-14B+VeriThinker | 91.0% | 0.973 | 0.922 | 0.947 | 95.5% | 0.986 | 0.966 | 0.976 |
| R1-Distill-Llama-8B+VeriThinker | 87.4% | 0.972 | 0.880 | 0.924 | 93.9% | 0.981 | 0.955 | 0.968 |

Table 3: Speculative Reasoning Results. We applied VeriThinker to three LRMs and employed two short-CoT LLMs as draft models to evaluate the performance of solution-wise speculative reasoning. We report accuracy, token counts, throughput, and AcR (long-CoT reasoning Activation Rate). Underlined numbers indicate the token counts during the draft phase for short-CoT LLMs.

| Method | MATH500 | | | | GSM8K | | | |
|---|---|---|---|---|---|---|---|---|
| | Acc ↑ | Tokens ↓ | Throughput ↑ | AcR ↓ | Acc ↑ | Tokens ↓ | Throughput ↑ | AcR ↓ |
| R1-Distill-Qwen-7B | 94.0% | 3791 | 1.0x | 100.0% | 92.8% | 1555 | 1.0x | 100.0% |
| +Qwen-2.5-Math-1.5B-Instruct | 91.8% | 589+889 | **4.4x** | 20.8% | 93.3% | 321+229 | **5.1x** | 14.0% |
| +Qwen-2.5-Math-7B-Instruct | 93.4% | 614+656 | **4.2x** | 14.6% | 96.1% | 294+113 | **7.1x** | 5.0% |
| R1-Distill-Qwen-14B | 95.2% | 3529 | 1.0x | 100.0% | 95.6% | 1309 | 1.0x | 100.0% |
| +Qwen-2.5-Math-1.5B-Instruct | 93.0% | 589+1130 | **3.6x** | 22.8% | 95.1% | 321+175 | **5.3x** | 13.2% |
| +Qwen-2.5-Math-7B-Instruct | 95.7% | 614+954 | **3.5x** | 17.6% | 96.6% | 294+93 | **5.4x** | 5.5% |
| R1-Distill-Llama-8B | 90.6% | 4361 | 1.0x | 100.0% | 89.2% | 1643 | 1.0x | 100.0% |
| +Qwen-2.5-Math-1.5B-Instruct | 90.4% | 589+1589 | **3.3x** | 25.8% | 91.7% | 321+307 | **4.7x** | 14.5% |
| +Qwen-2.5-Math-7B-Instruct | 91.6% | 614+1265 | **3.2x** | 21.2% | 96.4% | 294+161 | **4.8x** | 6.0% |

reasoning models at a 0.2:0.8 ratio to generate concise reasoning chains on PRM12K and GSM8K. Only correct CoT solutions are retained, and the average length of these concise chains is about 60% of the original. Both SFT and our VeriThinker method are finetuned with same LoRA configuration.

## 4.2 Experimental Results and Analysis

**CoT Compression Results.** We present comprehensive experimental results on CoT compression in Table 1. Our VeriThinker significantly reduces the number of tokens while maintaining or even slightly improving reasoning accuracy. In contrast, other baseline methods fail to achieve a good trade-off between efficiency and accuracy. When the accuracy is successfully maintained, the length of the reasoning chain shows little variation; when the token count is significantly reduced, the reasoning accuracy inevitably decreases substantially - this phenomenon is particularly evident in challenging problems. Among the baselines, the SFT method using target chain as optimization objective, while significantly reducing token count, leads to notable degradation in the model's self-reflection capability. Consequently, some incorrect steps fail to be double-checked, resulting in erroneous outcomes. Our VeriThinker, by contrast, enables the model to distinguish between correct and incorrect solutions, enhancing its ability to judge when self-checking is needed, thereby maximally preserving reasoning accuracy with even slight improvements.

As shown in the Table 1, our method demonstrates consistent superior performance across all three LLMs, with particularly outstanding results on the challenging AIME dataset. Our approach reduces the CoT length by approximately 29% (from 13,108 to 9,381 tokens) for the R1-Distill-Qwen-7B model on AIME 2024 while increasing reasoning accuracy by 2.4%. On AIME 2025, the token count decreases by about 28% (from 14,321 to 10,287) with a 2.1% accuracy improvement. We also observe that our method shows higher sensitivity to CoT compression on LLaMA architecture models. Although VeriThinker maintains good accuracy preservation on R1-Distill-LLaMA-8B, the compression ratio of reasoning chains is relatively smaller. These comprehensive experimental results strongly support and highlight the contributions of our method.

**CoT Correctness Verification Results.** As VeriThinker trains LLMs to verify the correctness of CoT solutions, we conducted additional experiments to analyze its verification accuracy. Specifically, we generated CoT solutions using Qwen-2.5-Math-1.5B-Instruct and Qwen-2.5-Math-7B-Instruct on GSM8K and Math500 datasets. We then employed SVFT LRMs to evaluate the correctness of these

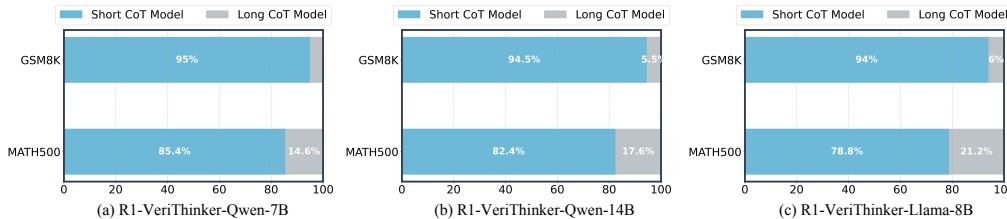

Figure 3: Speculative reasoning results on three reasoning models. When using Qwen-2.5-Math-Instruct-7B as the draft model, most problems in MATH500 and GSM8K can be solved with short CoT model, while only a few require activation of the long CoT model for more complex solutions.

solutions. As shown in Table 2, all three SVFT LRMs demonstrate high verification accuracy. The verification accuracy on GSM8K is higher than on MATH500, indicating that more concise and clear solutions are easier to classify correctly. We also observe a positive correlation between the model's verification accuracy and its reasoning capability. For reasoning tasks, the performance ranking is R1-Distill-14B > R1-Distill-7B > R1-Distill-8B, and the verification accuracy follows exactly the same trend. Furthermore, we find that LLMs generally exhibit notably higher precision than recall. This suggests that LLMs adopt a conservative approach when judging solution correctness - they rarely misclassify incorrect solutions as correct ones.

**Speculative Reasoning Results.** The above experiments demonstrate that SVFT LRMs achieve high verification accuracy, particularly in precision. Building on this strength, we conduct extensive experiments to assess their performance in speculative reasoning tasks. We employ Qwen-2.5-Math-1.5B-Instruct and Qwen-2.5-Math-7B-Instruct as draft models to observe the performance of SVFT-enhanced R1-Distill-7B, R1-Distill-8B, and R1-Distill-14B on solution-wise speculative reasoning. In addition to accuracy and average token count, we report throughput (the time cost for completing all problems in the dataset) and AcR (the activation ratio of LRM), which indicates the proportion of problems that activate the LRM's long-CoT reasoning. As shown in Table 3, speculative reasoning significantly improves the LRM's throughput while maintaining or even enhancing reasoning accuracy. This is attributed to its remarkably low AcR. For instance, as presented in Figure 3, when using Qwen-2.5-Math-7B-Instruct as the draft model, R1-Distill-7B only needs to conduct its own reasoning for 14.6% of problems in MATH500 and 5% in GSM8K. This implies that the draft model solves the majority of problems, while the LRM is only engaged for the few truly challenging problems requiring long-CoT reasoning, resulting in substantial efficiency gains. Speculative reasoning also boosts accuracy, particularly on GSM8K, because the LRM sometimes underperforms the short-CoT model on simple problems, and in these cases, it accepts the correct draft solution. The overall accuracy of speculative reasoning approximates the union of the draft model's and LRM's accuracies.

Unlike traditional speculative decoding, we observe that for speculative reasoning, a smaller draft model does not necessarily yield higher throughput. This is because the computational cost of short-CoT LLMs is negligible compared to the substantial inference cost of LRMs. For example, on the MATH500 dataset, Qwen-2.5-Math-7B-Instruct achieves approximately 20× the throughput of R1-Distill-7B. Thus, the primary factor affecting speedup is AcR: the fewer problems requiring LRM activation, the greater the throughput improvement. Larger draft models typically exhibit lower AcR due to their higher accuracy, thus not suffering throughput disadvantages compared to smaller draft models. However, this raises a potential issue: as problem difficulty increases, the throughput benefits of speculative reasoning may diminish.

**Applied to Short-CoT LLM.** We also applied VeriThinker to Qwen-2.5-Math-Instruct and Qwen-2.5-Instruct models to explore the effect of SVFT on short-CoT LLMs. As shown in Table 4, since short-CoT does not suffer from overthinking, our method does not significantly alter the average token count. However, SVFT improves reasoning accuracy on both MATH500 and GSM8K datasets, indicating that learning to verify helps the LLM better focus on key elements that influence reasoning correctness, thereby enhancing reasoning performance.

Table 4: Results on Short-CoT LLMs. We apply VeriThinker to three short-CoT models and evaluate their reasoning performance.

| Method | MATH500 | | GSM8K | |
|---|---|---|---|---|
| | Tokens ↓ | Acc. ↑ | Tokens ↓ | Acc. ↑ |
| Qwen-2.5-Math-7B-Instruct | 700 | 85.0% | 319 | 95.6% |
| + **VeriThinker** | 614 | **87.6%** | 304 | **96.7%** |
| Qwen-2.5-Math-1.5B-Instruct | 568 | 77.0% | 314 | 85.6% |
| + **VeriThinker** | 589 | **80.2%** | 321 | **86.1%** |
| Qwen-2.5-7B-Instruct | 635 | 78.6% | 294 | 91.8% |
| + **VeriThinker** | 577 | **80.2%** | 285 | **92.4%** |

## 5 Limitations

The primary limitation of our approach is that it does not effectively enable CoT compression in smaller models (e.g., a 1.5B-parameter reasoning model). This is largely due to our fine-tuning strategy: instead of optimizing directly on the original reasoning task, we fine-tune the model on CoT verification as an auxiliary task. Since smaller models have limited capacity, they are more prone to catastrophic forgetting during SVFT. As a result, they struggle to maintain high reasoning accuracy while performing effective CoT compression.

## 6 Social Impacts

In this paper, we propose VeriThinker, a simple yet effective method for compressing reasoning chains while preserving high accuracy. As large reasoning models become increasingly prevalent, their lengthy reasoning chains lead to dramatically higher inference costs, hindering efficient deployment and limiting practical utility. Our method addresses this critical challenge by significantly reducing token usage without compromising reasoning performance, thereby enhancing the real-world applicability of reasoning models.

## 7 Conclusion

In this paper, we introduce VeriThinker, a novel approach for CoT compression. Our key innovation is a new fine-tuning method called supervised verification fine-tuning. For the first time, we fine-tune the LRM on an auxiliary verification task instead of the reasoning task itself to achieve effective CoT compression. This approach eliminates the dependency on target chain data, which can be difficult and expensive to obtain. We have performed an in-depth analysis to explain the underlying principles of the proposed methods. Extensive experiments demonstrate that our method significantly shortens the reasoning chain while preserving accuracy, even on highly challenging problems. Additionally, VeriThinker can be generalized to speculative decoding, achieving substantially higher throughput.

### Ackonwledgement

This project is supported by the National Research Foundation, Singapore, and Cyber Security Agency of Singapore under its National Cybersecurity R&D Programme and CyberSG R&D Cyber Research Programme Office (Award: CRPO-GC1-NTU-002).

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

## A    Training Details

We provide additional training details in this section. We employ *Low-Rank Adaptation* (LoRA) [18] for efficient fine-tuning, which significantly enhances training efficiency and effectively mitigates catastrophic forgetting, as training and inference are performed on different tasks.

Our LoRA configurations are presented in Table 5. We utilized different LoRA ranks and alpha values for the three distinct models to achieve the optimal balance between underfitting and catastrophic forgetting. All other training hyperparameters remain consistent across models: learning rate = 3e-5, LoRA dropout = 0.05, weight decay = 0.01, and batch size = 64. All models were trained for 2 epochs on our self-constructed CoT-Verification dataset.

Table 5: The LoRA configuration in our training process.

| Models | LoRA Module | LoraLoRARank | LoRA Alpha |
|---|---|---|---|
| DeepSeek-R1-Distill-Qwen-7B | QKVO | 256 | 512 |
| DeepSeek-R1-Distill-Qwen-14B | QKVO | 128 | 128 |
| DeepSeek-R1-Distill-Llama-8B | QKVO | 128 | 128 |

## B    Construct the CoT-Verification Dataset

As SVFT trains the LLM to directly distinguish whether a CoT solution is correct, a crucial challenge lies in constructing the CoT verification dataset for SVFT.

**Problem Collection.**  The first step involves collecting problems for our dataset. To ensure a diverse range of topics and difficulty levels, we aggregate problems from four mathematical datasets known for their breadth of content and varying difficulties: PRM12K [32], GSM8K [6], LIMO [70], and Numina-Math [29]. Specifically, we extract all problems from the training set of PRM12K, GSM8K, and LIMO. For Numina-Math, we only select problems whose solutions are integer-valued math-word-problems , simplifying correctness labeling. Following this procedure, we collected approximately 300K mathematical problems spanning diverse topics and difficulty levels.

**CoT Solution Collection.**    The second step entails generating numerous CoT solutions for the collected problems, which will serve as targets for our verification task. To achieve efficiency and variety, we employ several small, non-reasoning language models: `Qwen-2.5-0.5B-Instruct`, `Qwen-2.5-1.5B-Instruct`, `Qwen-2.5-7B-Instruct`, `Qwen-Math-1.5B-Instruct`, and `Qwen-Math-7B-Instruct` [66]. We utilize a CoT prompting strategy to generate step-by-step solutions for the previously collected 300K problems. Notably, none of our CoT solutions are generated by reasoning models, and none include explicit self-reflection steps. Experiments indicate that reasoning models producing long-chain solutions complicate optimization in SVFT, as a single response might contain multiple correct or incorrect sub-steps. This collection method offers two main advantages:

- Computational efficiency: All selected models are lightweight and only generating short-CoT solutions, significantly accelerating the data generation process. For instance, using `Qwen-Math-1.5B-Instruct`, we generate solutions for all 300K problems within 3 to 4 hours using only four NVIDIA A6000 GPUs.

- Solution diversity: The selected models exhibit varying reasoning capabilities, producing a diverse set of correct and incorrect solutions for each problem. This diversity greatly enhances the robustness of subsequent fine-tuning.

Consequently, we obtain five different short CoT solutions per problem, yielding a total of 1.5 million problem-solution pairs.

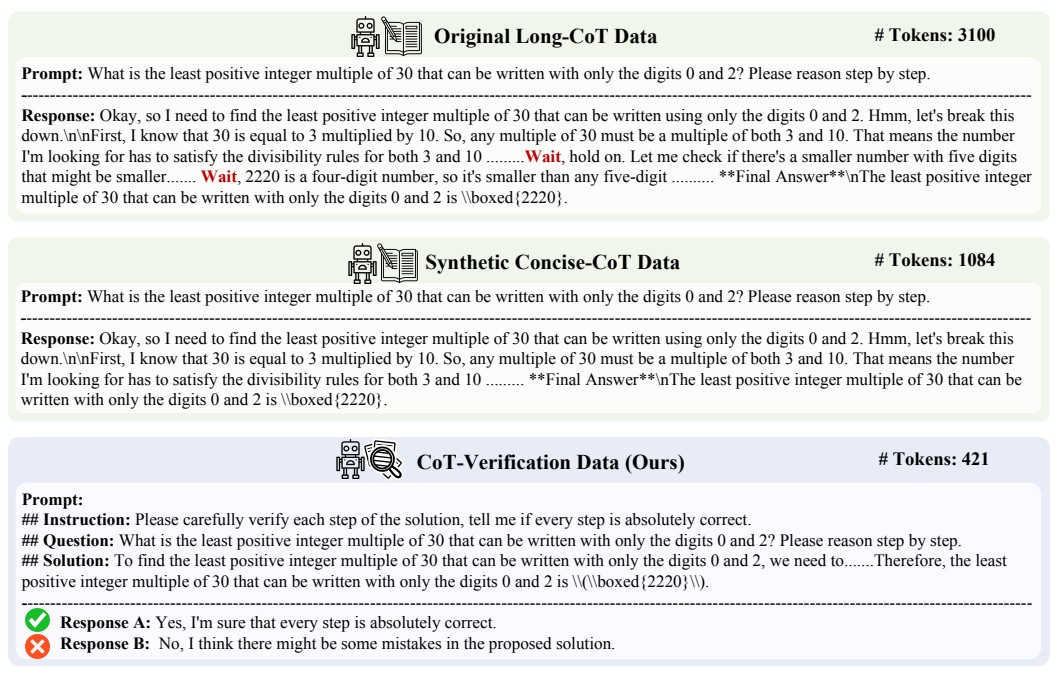

Figure 4: Training data format comparison.

**Correctness Labeling.** In the third step, we label the correctness of each generated CoT solution. Rather than evaluating each reasoning step, we simplify labeling by verifying only the final answers against known ground-truth solutions. As all collected problems have deterministic solutions, correctness labeling is straightforward: we employ the Hugging Face `math_verify` function to automatically extract final answers from CoT solutions and compare them against ground truths. Following this procedure, each of the 1.5 million problem-solution pairs is labeled as either correct or incorrect.

**Verification Data Selection.** Finally, we select a subset from the 1.5 million labeled pairs for fine-tuning. Initially, we discard problems where all five CoT solutions are uniformly correct or incorrect. Such problems lack informative training signals due to being either too trivial or excessively challenging, and this process also helps filter out inherently problematic data. Next, we apply a straightforward deduplication strategy using `Qwen-Math-1.5B-Instruct` as a reference model. Specifically, for problems correctly solved by the reference model, we retain its correct solutions along with incorrect solutions generated by the other models. Conversely, for problems incorrectly solved by the reference model, we retain its incorrect solutions and also incorrect solutions from the other models. This selection strategy ensures each problem contributes at least one correct and one incorrect CoT solution. Ultimately, this process yields a fine-tuning dataset consisting of 350K instances, comprising approximately 160K correct and 190K incorrect CoT solutions. Each instance is reformatted according to the structure illustrated in Figure 1.

## C   Cross-Domain Generalization

Our proposed VeriThinker method demonstrates strong generalization ability. Even when there exists a domain gap between the verification data used for training and the reasoning tasks during inference, our method continues to perform effectively. To substantiate this claim, we conduct two additional experiments.

We first construct a small CoT verification dataset using only questions from PRM12K and GSM8K, and fine-tune our reasoning model on this restricted data. We then evaluate the model on the much more challenging AIME datasets. As shown in Table 6, even with training data limited to PRM12K and GSM8K, our method significantly reduces token usage while maintaining comparable accuracy on the AIME benchmarks.

Table 6: Fine-tuned on PRM12K and GSM8K, evaluated on the more difficult AIME datasets.

| Model | MATH500 | AIME24 | AIME25 |
|---|---|---|---|
| R1-Distill-7B | 94.0% (3791 tokens) | 54.1% (13108 tokens) | 38.7% (14321 tokens) |
| **+VeriThinker** | **95.0% (2534 tokens)** | **56.4% (10302 tokens)** | **39.8% (11270 tokens)** |
| R1-Distill-14B | 95.2% (3529 tokens) | 69.0% (11724 tokens) | 49.6% (13409 tokens) |
| **+VeriThinker** | **95.2% (2505 tokens)** | **68.7% (8425 tokens)** | **49.2% (10580 tokens)** |

To further illustrate this generalization ability, we evaluate our model, trained exclusively on mathematical reasoning tasks, on the GPQA-Diamond dataset [48]. This benchmark consists of multiple-choice questions spanning diverse scientific domains (e.g., chemistry, physics, biology), and differs substantially in both format and content from our training data. As shown in Table 7, our approach consistently compresses the length of CoTs while maintaining accuracy.

Table 7: Cross-domain performance on GPQA-Diamond.

| Model | GPQA-Diamond |
|---|---|
| R1-Distill-7B | 51.0% (7407 tokens) |
| **+VeriThinker** | **50.0% (4984 tokens)** |
| R1-Distill-14B | 60.1% (6996 tokens) |
| **+VeriThinker** | **60.6% (4309 tokens)** |

Overall, these results demonstrate that VeriThinker achieves robust performance even in the presence of substantial domain shifts between training and testing, confirming its strong cross-domain generalization capability.

## D  Comparison of Verification Capability

To further validate our approach, we use Qwen-2.5-1.5B-Instruct to generate CoT solutions on MATH500 and evaluated the verification accuracy of several models. We compared our VeriThinker-7B with two strong open-source models (Qwen-2.5-7B-Instruct and LLaMA-3.1-8B-Instruct) and two advanced closed-source models (OpenAI GPT-4o and GPT-4.1). As shown in Table 8, our method achieves substantially higher verification accuracy than the open-source models and even surpasses the two large closed-source models from OpenAI.

Table 8: Comparison of CoT correctness verification capability.

| Model | Accuracy (%) | Precision | Recall | F1 |
|---|---|---|---|---|
| Qwen-2.5-7B-Instruct | 74.8 | 0.714 | 0.934 | 0.810 |
| LLaMA-3.1-8B-Instruct | 70.0 | 0.668 | **0.948** | 0.783 |
| OpenAI GPT-4o | 85.0 | 0.889 | 0.843 | 0.866 |
| OpenAI GPT-4.1 | 86.4 | 0.910 | 0.847 | 0.877 |
| **VeriThinker-7B (Ours)** | **90.4** | **0.972** | 0.857 | **0.911** |

## E  Case Study

We also present additional case studies on cot compression in Figure 5 and Figure 6.

# CoT Compression Case Study

**Prompt:** The set of points $(x,y,z)$ that satisfy \n\[2x = 3y = -z\]\is a line. \n\nThe set of points $(x,y,z)$ that satisfy \n\[6x = -y = -4z\]\is another line. \n\nFind the angle between these lines, in degrees.

---------------------------------------------------------------------------------------------------------------------------------

**Original Response:** Okay, so I need to find the angle between two lines in 3D space. The lines are given by the equations:\n\nFirst line: \\(2x = 3y = -z\\)\n\nSecond line: \\(6x = -y = -4z\\)\n\nHmm, I remember that to find the angle between two lines, I can use the dot product formula involving their direction vectors. So, first, I need to find the direction vectors of both lines.\n\nLet me start with the first line: \\(2x = 3y = -z\\). I think this is a standard symmetric equation of a line. The general form is \\(\\frac{x - x_0}{a} = \\frac{y - y_0}{b} = \\frac{z - z_0}{c}\\), where \\((x_0, y_0, z_0)\\) is a point on the line and \\((a, b, c)\\) is the direction vector. Looking at the first line, \\(2x = 3y = -z\\), I can set each of these equal to a parameter, let's say \\(k\\). So,\n\n\\(2x = k\\), which implies \\(x = \\frac{k}{2}\\)\n\n\\(3y = k\\), which implies \\(y = \\frac{k}{3}\\)\n\n\\(-z = k\\), which implies \\(z = -k\\)\n\nSo, if I write the parametric equations:\n\n\\(x = \\frac{k}{2}\\)\n\n\\(y = \\frac{k}{3}\\)\n\n\\(z = -k\\)\n\nTherefore, the direction vector for the first line can be found by looking at the coefficients of \\(k\\) in each parametric equation. So, direction vector \\( \\vec{v_1} = \\left( \\frac{1}{2}, \\frac{1}{3}, -1 \\right) \\).\n\nWait, but direction vectors are usually given in integers, so maybe I can scale this vector to eliminate the fractions. If I multiply each component by 6, which is the least common multiple of 2 and 3, I get \\( \\vec{v_1} = (3, 2, -6) \\). That should work because scaling a vector doesn't change its direction.\n\nOkay, moving on to the second line: \\(6x = -y = -4z\\). I'll do the same process.\n\nLet me set each part equal to another parameter, say \\(m\\):\n\n\\(6x = m\\) → \\(x = \\frac{m}{6}\\)\n\n\\(-y = m\\) → \\(y = -m\\)\n\n\\(-4z = m\\) → \\(z = -\\frac{m}{4}\\)\n\nSo, parametric equations: \\(x = \\frac{m}{6}\\)\n\n\\(y = -m\\)\n\n\\(z = -\\frac{m}{4}\\)\n\nTherefore, the direction vector \\( \\vec{v_2} = \\left( \\frac{1}{6}, -1, -\\frac{1}{4} \\right) \\). Again, maybe I should scale this vector to make it simpler. The denominators are 6, 1, and 4, so the least common multiple is 12. Multiply each component by 12:\n\n\\( \\vec{v_2} = (2, -12, -3) \\). Hmm, let me check:\n\n12*(1/6) = 2, 12*(-1) = -12, 12*(-1/4) = -3. Yeah, that works.\n\nWait, actually, 12 is the LCM of 6, 1, and 4? Let me confirm:\n\n6 factors into 2*3, 1 is 1, and 4 is 2². So LCM is 2²*3 = 12. Yeah, that's correct. So scaling by 12 gives the direction vector \\( (2, -12, -3) \\).\n\nAlternatively, maybe I could have found the direction vector differently. Let me see:\n\nFrom the symmetric equations, the direction ratios are the denominators. So for the second line, \\(6x = -y = -4z\\), if I write it as \\(\\frac{x}{1/6} = \\frac{y}{-1} = \\frac{z}{-1/4}\\), so direction vector is \\( (1/6, -1, -1/4) \\), which is what I had before, just scaled. So scaling up gives \\( (2, -12, -3) \\). Okay, that seems consistent.\n\nNow, I have two direction vectors:\n\n\\( \\vec{v_1} = (3, 2, -6) \\)\n\n\\( \\vec{v_2} = (2, -12, -3) \\)\n\nTo find the angle between the two lines, I can use the dot product formula:\n\n\\( \\vec{v_1} \\cdot \\vec{v_2} = |\\vec{v_1}| |\\vec{v_2}| \\cos \\theta \\) \n\nSo, solving for \\( \\theta \\):\n\n\\( \\cos \\theta = \\frac{ \\vec{v_1} \\cdot \\vec{v_2} }{ |\\vec{v_1}| |\\vec{v_2}| } \\)\n\nFirst, compute the dot product:\n\n\\( \\vec{v_1} \\cdot \\vec{v_2} = (3)(2) + (2)(-12) + (-6)(-3) \\)\n\nCalculate each term:\n\n3*2 = 6\n\n2*(-12) = -24\n\n-6*(-3) = 18\n\nAdd them up: 6 - 24 + 18 = (6 + 18) - 24 = 24 - 24 = 0\n\nWait, the dot product is zero? That means the vectors are perpendicular. So, does that mean the angle between the lines is 90 degrees?\n\nBut let me double-check my calculations because sometimes scaling can affect things.\n\nWait, no, the direction vectors are just scaled versions of the original vectors, so their dot product being zero implies that the original vectors are also perpendicular. So the angle is 90 degrees.\n\nBut just to make sure, let me compute the magnitudes as well.\n\nCompute \\( |\\vec{v_1}| \\):\n\n\\( |\\vec{v_1}| = \\sqrt{3^2 + 2^2 + (-6)^2} = \\sqrt{9 + 4 + 36} = \\sqrt{49} = 7 \\)\n\nCompute \\( |\\vec{v_2}| \\):\n\n\\( |\\vec{v_2}| = \\sqrt{2^2 + (-12)^2 + (-3)^2} = \\sqrt{4 + 144 + 9} = \\sqrt{157} \\)\n\nSo, then, the dot product is 0, so \\( \\cos \\theta = 0 \\), which means \\( \\theta = 90^\\circ \\).\n\nWait, but just to make sure, is that correct? Because sometimes, depending on how you parameterize, you might get a different direction vector, but in this case, both direction vectors were scaled appropriately, so the angle should be the same.\n\nAlternatively, let's think about the initial direction vectors before scaling.\n\nOriginal direction vectors:\n\n\\( \\vec{v_1} = (3, 2, -6) \\)\n\n\\( \\vec{v_2} = (2, -12, -3) \\)\n\nDot product:\n\n3*2 + 2*(-12) + (-6)*(-3) = 6 -24 +18 = 0\n\nSame result. So, yeah, the angle is 90 degrees.\n\nBut wait, let me visualize this. If two lines have direction vectors that are perpendicular, then the lines themselves are perpendicular. So, that seems right.\n\nAlternatively, maybe I can compute the angle using the original direction vectors before scaling:\n\n\\( \\vec{v_1} = (3, 2, -6) \\), \\( \\vec{v_2} = (2, -12, -3) \\)\n\nDot product is 0, as before. So, yeah, the angle is 90 degrees.\n\nWait, but just to be thorough, let me compute the angle step by step.\n\nCompute \\( \\vec{v_1} \\cdot \\vec{v_2} = 3*2 + 2*(-12) + (-6)*(-3) = 6 -24 +18 = 0 \\)\n\nCompute \\( |\\vec{v_1}| = \\sqrt{3^2 + 2^2 + (-6)^2} = \\sqrt{9 + 4 + 36} = \\sqrt{49} =7 \\)\n\nCompute \\( |\\vec{v_2}| = \\sqrt{2^2 + (-12)^2 + (-3)^2} = \\sqrt{4 + 144 + 9} = \\sqrt{157} \\)\n\nSo, \\( \\cos \\theta = 0 / (7 * \\sqrt{157}) = 0 \\)\n\nThus, \\( \\theta = \\arccos(0) = 90^\\circ \\)\n\nTherefore, the angle between the two lines is 90 degrees.\n\nWait, that seems a bit surprising. Let me think again.\n\nAlternatively, maybe I made a mistake in computing the direction vectors. Let me double-check.\n\nFirst line: \\(2x = 3y = -z\\). Let me write it as \\(2x = 3y = -z = k\\). Then:\n\nx = k/2\n\ny = k/3\n\nz = -k\n\nTherefore, direction vector is (1/2, 1/3, -1). To eliminate denominators, multiply by 6, getting (3, 2, -6). That seems correct.\n\nSecond line: \\(6x = -y = -4z\\). Let me set each equal to m:\n\n6x = m → x = m/6\n\n-y = m → y = -m\n\n-4z = m → z = -m/4\n\nTherefore, direction vector is (1/6, -1, -1/4). Multiply by 12 to eliminate denominators: (2, -12, -3). That also seems correct.\n\nDot product: (3)(2) + (2)(-12) + (-6)(-3) = 6 -24 +18 = 0. So, correct.\n\nSo, the angle is indeed 90 degrees. So, the two lines are perpendicular.\n\nTherefore, the angle between them is 90 degrees.\n\n\n**Final Answer**\nThe angle between the lines is \\boxed{90} degrees.

**# Tokens: 2330**

---------------------------------------------------------------------------------------------------------------------------------

**Our Response:** Okay, so I need to find the angle between two lines in 3D space. The lines are given by the equations:\n\nFirst line: \\(2x = 3y = -z\\)\n\nSecond line: \\(6x = -y = -4z\\)\n\nHmm, I remember that to find the angle between two lines, I can use the direction vectors of the lines and then compute the angle between those vectors using the dot product formula. So, first, I need to find the direction vectors for both lines.\n\nLet me start with the first line: \\(2x = 3y = -z\\). I think this can be written in parametric form by setting each expression equal to a parameter, say \\(t\\). So, let me set \\(2x = 3y = -z = t\\). Then, I can express \\(x\\), \\(y\\), and \\(z\\) in terms of \\(t\\):\n\n- \\(2x = t \\Rightarrow x = t/2\\)\n- \\(3y = t \\Rightarrow y = t/3\\)\n- \\(-z = t \\Rightarrow z = -t\\)\n\nSo, the parametric equations are:\n\n\\(x = \\frac{t}{2}\\),\n\n\\(y = \\frac{t}{3}\\),\n\n\\(z = -t\\).\n\nTherefore, the direction vector for the first line is given by the coefficients of \\(t\\) in each component, which is \\(\\left(\\frac{1}{2}, \\frac{1}{3}, -1\\right)\\). But usually, direction vectors are given as integer vectors, so I can multiply by 6 to eliminate the fractions:\n\n\\(6 \\times \\left(\\frac{1}{2}, \\frac{1}{3}, -1\\right) = (3, 2, -6)\\). So, the direction vector for the first line is \\(\\mathbf{v} = (3, 2, -6)\\).\n\nNow, moving on to the second line: \\(6x = -y = -4z\\). Similarly, I can set each expression equal to a parameter, say \\(s\\). So, let me set \\(6x = -y = -4z = s\\). Then, express each variable in terms of \\(s\\):\n\n- \\(6x = s \\Rightarrow x = s/6\\)\n- \\(-y = s \\Rightarrow y = -s\\)\n- \\(-4z = s \\Rightarrow z = -s/4\\)\n\nSo, the parametric equations are:\n\n\\(x = \\frac{s}{6}\\),\n\n\\(y = -s\\),\n\n\\(z = -\\frac{s}{4}\\).\n\nTherefore, the direction vector is given by the coefficients of \\(s\\), which is \\(\\left(\\frac{1}{6}, -1, -\\frac{1}{4}\\right)\\). Again, to make it an integer vector, I can find a common denominator. The denominators are 6, 1, and 4, so the least common multiple is 12. Multiply each component by 12:\n\n\\(12 \\times \\left(\\frac{1}{6}, -1, -\\frac{1}{4}\\right) = (2, -12, -3)\\). So, the direction vector for the second line is \\(\\mathbf{w} = (2, -12, -3)\\).\n\nNow, I have the two direction vectors:\n\n\\(\\mathbf{v} = (3, 2, -6)\\) and \\(\\mathbf{w} = (2, -12, -3)\\).\n\nThe formula for the angle \\(\\theta\\) between two vectors is:\n\n\\(\\cos \\theta = \\frac{\\mathbf{v} \\cdot \\mathbf{w}}{|\\mathbf{v}| |\\mathbf{w}|}\\)\n\nFirst, let's compute the dot product \\(\\mathbf{v} \\cdot \\mathbf{w}\\):\n\n\\3 \\times 2 + 2 \\times (-12) + (-6) \\times (-3) = 6 - 24 + 18 = 0\n\nWait, the dot product is 0? That means the vectors are perpendicular, so the angle between them is 90 degrees. Hmm, that seems surprising, but let me double-check my calculations.\n\nCalculating again:\n- \\(3 \\times 2 = 6\\)\n- \\(2 \\times (-12) = -24\\)\n- \\(-6 \\times (-3) = 18\\)\n\nAdding them up: \\(6 - 24 + 18 = 0\\). Yes, that's correct. So the dot product is indeed 0, which means the vectors are orthogonal.\n\nTherefore, the angle between the lines is 90 degrees.\n\nBut just to be thorough, let me compute the magnitudes of the vectors as well to ensure I didn't make a mistake there.\n\nCompute \\(|\\mathbf{v}|\\):\n\n\\(|\\mathbf{v}| = \\sqrt{3^2 + 2^2 + (-6)^2} = \\sqrt{9 + 4 + 36} = \\sqrt{49} = 7\\)\n\nCompute \\(|\\mathbf{w}|\\):\n\n\\(|\\mathbf{w}| = \\sqrt{2^2 + (-12)^2 + (-3)^2} = \\sqrt{4 + 144 + 9} = \\sqrt{157} \\approx 12.53\\)\n\nBut since the dot product is 0, the angle is 90 degrees regardless of the magnitudes. So, yeah, I can confidently say that the angle between the two lines is 90 degrees.\n\n\n**Final Answer**\nThe angle between the lines is \\boxed{90} degrees.

**# Tokens: 1267**

Figure 5: Case study 2 on CoT Compression.

## CoT Compression Case Study

**Prompt:** Evaluate $(1+2i)6-3i$.

---

**Original Response:** Okay, so I need to evaluate the expression $(1 + 2i)6 - 3i$. Hmm, let me think about how to approach this. I remember that when dealing with complex numbers, we can perform operations like addition, subtraction, multiplication, and division. In this case, it seems like we have a complex number multiplied by 6 and then subtracting 3i. Let me break it down step by step.\n\nFirst, I need to multiply the complex number $(1 + 2i)$ by 6. I think multiplying a complex number by a real number is straightforward. You just distribute the real number to both the real and imaginary parts of the complex number. So, 6 times 1 is 6, and 6 times 2i is 12i. Therefore, $(1 + 2i)$ multiplied by 6 should give me 6 + 12i.\n\nLet me write that down:\n\n$(1 + 2i) * 6 = 6 * 1 + 6 * 2i = 6 + 12i$.\n\nOkay, so the first part is 6 + 12i. Now, the expression is this result minus 3i. So, I need to subtract 3i from 6 + 12i. Let me do that.\n\nSubtracting 3i from 6 + 12i is the same as adding -3i to 6 + 12i. So, I can write that as:\n\n6 + 12i - 3i.\n\nNow, I can combine the like terms, which are the imaginary parts. So, 12i minus 3i is... let me calculate that. 12 minus 3 is 9, so that should be 9i.\n\nPutting it all together, the real part is 6, and the imaginary part is 9i. Therefore, the result is 6 + 9i.\n\nWait, let me make sure I didn't make a mistake. So, starting with $(1 + 2i) * 6$. Multiplying each term:\n\n1 * 6 = 6,\n\n2i * 6 = 12i.\n\nSo, that gives 6 + 12i. Then subtracting 3i: 6 + 12i - 3i. 12i - 3i is indeed 9i, so 6 + 9i. That seems correct.\n\nI guess another way to think about it is to factor out the 3i or something, but in this case, since it's just subtracting 3i, combining like terms is straightforward.\n\nLet me also consider if there was a different interpretation of the original expression. It could be interpreted as $(1 + 2i)$ multiplied by $(6 - 3i)$, but the original expression is $(1 + 2i)6 - 3i$, which is written with the 6 right after the parenthesis, so I think it's 6 times $(1 + 2i)$, then subtract 3i.\n\nBut just to be thorough, what if it was $(1 + 2i)(6 - 3i)$? How would that be different? Let me calculate that for fun.\n\nUsing the distributive property, $(1)(6) + (1)(-3i) + (2i)(6) + (2i)(-3i)$.\n\nCalculating each term:\n\n1 * 6 = 6,\n\n1 * (-3i) = -3i,\n\n2i * 6 = 12i,\n\n2i * (-3i) = -6i^2.\n\nNow, combine the terms:\n\n6 - 3i + 12i - 6i^2.\n\nSimplify like terms: -3i + 12i is 9i.\n\nSo, 6 + 9i - 6i^2.\n\nBut wait, i^2 is equal to -1, so -6i^2 is -6*(-1) which is +6.\n\nSo, 6 + 6 + 9i = 12 + 9i.\n\nHmm, so if the original expression was $(1 + 2i)(6 - 3i)$, the result would be 12 + 9i. But in our problem, the original expression is $(1 + 2i)6 - 3i$, which is 6*$(1 + 2i)$ - 3i, resulting in 6 + 9i. So, the initial interpretation was correct.\n\nTherefore, I think my answer is 6 + 9i.\n\nJust to recap, the steps are:\n\n1. Multiply $(1 + 2i)$ by 6: 6*1 + 6*2i = 6 + 12i.\n\n2. Subtract 3i from the result: 6 + 12i - 3i = 6 + 9i.\n\nYes, that seems correct. I don't think I made any mistakes here.\n\nAlternatively, if I were to represent this in the form a + bi, 6 is the real part and 9 is the imaginary coefficient, so the expression is simplified as 6 + 9i.\n\nLet me also consider if I can represent this in polar form, but since the question just asks to evaluate it, probably Cartesian form is sufficient.\n\nBut just for practice, let me convert 6 + 9i to polar form.\n\nFirst, find the modulus: sqrt(6^2 + 9^2) = sqrt(36 + 81) = sqrt(117). Hmm, sqrt(117) can be simplified as sqrt(9*13) = 3*sqrt(13). So, modulus is 3√13.\n\nThen, the argument θ is arctangent of (9/6) which is arctangent(3/2). So, θ = arctan(3/2). That's approximately 56.31 degrees, but we can leave it as arctan(3/2) radians.\n\nSo, in polar form, it's 3√13 (cos(arctan(3/2)) + i sin(arctan(3/2))). But I don't think the question requires this, so I can stick with the Cartesian form.\n\nSo, after evaluating $(1 + 2i)6 - 3i$, the result is 6 + 9i.\n\nI think that's the final answer.\n\n**Final Answer**\n\nThe result of evaluating $((1 + 2i)6 - 3i)$ is $\boxed{6 + 9i}$.                    **# Tokens: 1406**

---

**Our Response:** Okay, so I need to evaluate the expression $(1 + 2i)6 - 3i$. Hmm, let me think about how to approach this. I remember that when dealing with complex numbers, you have to distribute and then combine like terms. So, first, I should probably handle the multiplication part, which is $(1 + 2i)$ multiplied by 6, and then subtract 3i from the result.\n\nLet me write that down step by step. The original expression is $(1 + 2i)6 - 3i$. I can rewrite this as 6*$(1 + 2i)$ - 3i. Now, distributing the 6 across the terms inside the parenthesis. So, 6*1 is 6, and 6*2i is 12i. That gives me 6 + 12i. Then, I need to subtract 3i from that.\n\nSo, putting it all together, 6 + 12i - 3i. Now, I can combine the like terms, which are the imaginary parts. 12i - 3i is 9i. Therefore, the expression simplifies to 6 + 9i.\n\nWait, let me make sure I did that correctly. So, 6*$(1 + 2i)$ is indeed 6 + 12i. Then subtracting 3i gives me 6 + 12i - 3i. Yes, combining 12i and -3i gives 9i. So, the final result is 6 + 9i.\n\nIs there anything else I need to do here? I don't think so. It seems straightforward. I just multiplied the complex number by 6 and then subtracted 3i. Let me double-check my multiplication to be sure. 6 times 1 is 6, that's correct. 6 times 2i is 12i, that's right. Then subtracting 3i gives 9i. So, yes, the answer should be 6 + 9i.\n\nI guess that's it. I don't see any mistakes in my calculations. So, I can confidently say that the evaluated expression is 6 + 9i.\n\n**Final Answer**\nThe value of the expression is $\boxed{6 + 9i}$.                    **# Tokens: 489**

Figure 6: Case study 3 on CoT Compression.