# OpenReview forum: "VeriThinker: Learning to Verify Makes Reasoning Model Efficient"
_NeurIPS.cc/2025/Conference — NeurIPS 2025 poster_

### Official Review · Reviewer_gTAt · 2025-06-03

**Clarity:** 3
**Significance:** 3
**Originality:** 3
**Rating:** 4
**Confidence:** 3

**Summary:**

This paper introduces VeriThinker, a method designed to mitigate the "overthinking" issue in Large Reasoning Models (LRMs) which leads to excessively long reasoning chains and high inference costs. The core idea is Supervised Verification Finetuning (SVFT), where an LRM is finetuned on an auxiliary task of verifying the correctness of Chain-of-Thought (CoT) solutions. The authors posit that by learning to accurately assess CoT solution correctness, the LRM becomes more adept at deciding when self-reflection is truly necessary, thereby compressing the reasoning chain while aiming to maintain or even improve accuracy. The paper presents experimental results on several mathematical reasoning benchmarks to support VeriThinker's efficacy in reducing token count and its potential in applications like solution-wise speculative reasoning (SSR) and for enhancing short CoT LLMs.

**Questions:**

See above

**Ethical Concerns:**

["NO or VERY MINOR ethics concerns only"]

**Final Justification:**

I had no major concerns about the core intuition or overall motivation of the paper. However, my primary issue was with the choice of baseline and the lack of clarity around the dataset construction, which I believe weaken the empirical evaluation. Reading other reviewers' comments further reinforced these concerns.

I did not raise expectations around model scale or training cost, considering the resource constraints common in the current research landscape. Given this, I believe my current borderline accept rating is already a fairly generous evaluation.

**Limitations:**

yes

**Quality:**

3

**Strengths And Weaknesses:**

**strengths**
- Intuitively, I think the proposed VeriThinker method, which uses an auxiliary verification task to help models discern when to stop or continue thinking, makes sense. The idea of training a model to verify solutions to indirectly improve its reasoning efficiency is quite an original approach to the "overthinking" problem in LRMs. The experimental results presented, such as significant token reduction on benchmarks like MATH500 and AIME while maintaining or even slightly improving accuracy, also look promising and suggest the method has practical value.

**weaknesess/questions**
- The paper mentions that the SFT baseline uses "synthesized concise CoT chains as targets", but I feel there's insufficient detail on how these concise CoTs were generated. If the quality of these synthetic CoTs for the SFT baseline was not particularly high, it might make VeriThinker's relative performance appear more favorable than it would be against a stronger, well-optimized SFT baseline. More transparency and better quality on the SFT target data generation and its finetuning setup would be beneficial for a more complete assessment of fairness.

---

> ### Author Rebuttal · Authors · 2025-07-30
>
> Thanks for the thoughtful and valuable feedback. Please find our responses below, we hope they adequately address your concerns. If you have any further questions or need additional clarification, we’d be happy to discuss them with you. Thanks again for your time and efforts.
>
> > **Q1: The paper mentions that the SFT baseline uses "synthesized concise CoT chains as targets", but I feel there's insufficient detail on how these concise CoTs were generated.**
>
> **A:** Thanks for the insightful comments. We would like to clarify this point.
>
> To synthesize concise reasoning chains for the SFT baseline, we follow CoT-Valve [1] by merging the base and reasoning models at a 0.2:0.8 ratio to generate concise reasoning chains on PRM12K and GSM8K. Only correct CoT solutions are retained, and the average length of these concise chains is about 60% of the original. Both SFT and our VeriThinker method are finetuned using LoRA with same configuration.
>
> To ensure a fair comparison, we also created a small verification dataset using only PRM12K and GSM8K problems and trained our SVFT method with this restricted data. As shown in Tables 1 and 2, our approach still outperforms SFT, especially on the more difficult AIME benchmark.
>
> It’s also important to highlight that generating data for SFT is much more expensive than for our method. SFT relies on a merged long-CoT model to produce training data. In contrast, our method only needs a short-CoT model to generate solutions for verification, making the process much more efficient. For example, using the vLLM framework on the same 4 × A5000 GPUs, a 7B short-CoT model achieves 10× higher throughput than the merged long-CoT model used for SFT.
>
> #### *Table 1 Performance comparison with SFT on the same training dataset (R1-Distill-7B).*
> | Method| MATH500 | AIME24 |AIME25 |
> |--|--|--|--|
> | R1-Distill-7B | 94.0% (3791 tokens)| 54.1% (13108 tokens) | 38.7% (14321 tokens)|
> | +SFT (PRM12K+GSM8K) | 91.2% (2064 tokens)| 49.1% (8843 tokens) | 32.1% (9686 tokens)|
> | +VeriThinker(PRM12K+GSM8K)| 95.0% (2534 tokens)| 56.4% (10402 tokens) | 39.8% (11270 tokens)|
> | +VeriThinker (Complete Verification Data) | 94.8% (2125 tokens)| 56.5% (9381 tokens) | 40.8% (10287 tokens)|
>
> #### *Table 2 Performance comparison with SFT on the same training dataset (R1-Distill-14B).*
> | Method| MATH500 | AIME24 |AIME25 |
> |--|--|--|--|
> | R1-Distill-14B | 95.2% (3529 tokens)| 69.0% (11724 tokens) | 49.6% (13409 tokens)|
> | +SFT (PRM12K+GSM8K) | 92.8% (2226 tokens)| 58.3% (9021 tokens) | 43.3% (10644 tokens)|
> | +VeriThinker (PRM12K+GSM8K)| 95.2% (2505 tokens)| 68.7% (8425 tokens) | 49.2% (10580 tokens)|
> | +VeriThinker (Complete Verification Data) | 95.0% (2255 tokens)| 73.0% (7423 tokens) | 54.8% (9304 tokens)|
>
> ### Reference
> [1] CoT-Valve: Length-Compressible Chain-of-Thought Tuning. In Proceedings of the 63rd Annual Meeting of the Association for Computational Linguistics.

---

> > ### Comment · Reviewer_gTAt · 2025-08-05
> >
> > Thank you to the authors for the detailed responses, especially the clarification regarding the SFT baseline construction. I understand the core motivation of VeriThinker, and I find the idea of improving reasoning efficiency through an auxiliary verification task to be intuitive. I still have some reservations about the strength of the empirical results. While the authors have explained their baseline setup, I noticed that the SFT baseline exhibits a substantial drop in performance, which seems inconsistent with prior claims in works such as CoT-Valve.
> >
> > Additionally, I agree with several points raised by other reviewers, including the limited model diversity and the lack of a more systematic analysis of the cost-versus-performance trade-off. I believe these aspects could be better addressed in future versions of the paper. For these reasons, I will maintain my borderline accept rating.

---

> > > ### Author Response · Authors · 2025-08-05
> > > **Thanks for acknowledging our rebuttal and for the thoughtful and constructive feedback.**
> > >
> > > Thanks for acknowledging our rebuttal and for the thoughtful and constructive feedback.
> > >
> > > We wish to respectfully clarify that our claim regarding the accuracy drop of the SFT method on more challenging problems is not in conflict with previous works. For example, as shown in Table 3 of the CoT-Valve paper (regarding R1-Distill-8B), CoT-Valve reduces the token usage on AIME24 by approximately 27% after SFT, but this comes at the cost of a significant 10% drop in accuracy. This result is consistent with our own claims.
> > >
> > > Regarding model diversity, in addition to the three reasoning models already included in our main paper, we have reported experimental results on the phi-4-mini-reasoning model in our rebuttal. Furthermore, we plan to include results for reasoning models with more than 30 B parameters in the next version of our paper.
> > >
> > > Once again, we sincerely thank you for your valuable comments, which have been extremely helpful to us. We are committed to further improving the quality of our work in the next revision.

---

### Official Review · Reviewer_c9kd · 2025-06-18

**Clarity:** 3
**Significance:** 3
**Originality:** 3
**Rating:** 5
**Confidence:** 4

**Summary:**

This paper considers the problem of overthinking by the reasoning LLMs. It proposes a method for compressing the chain of thoughts in reasoning LLMs. The idea is to “fine-tune the model solely through an auxiliary verification task” instead of the previous approaches where models are fine-tuned on the original reasoning task using synthetic concise CoT data. When the model is trained to verify the correctness of CoT solutions, it becomes more diligent about its own chain of thoughts, and ultimately becomes less prone to overthinking. The paper experiments on several reasoning tasks reporting that the proposed method reduces the computational cost and increases the accuracy. Three models are considered: DeepSeek-R1-Distill-Qwen-7B, 57 DeepSeek-R1-Distill-Qwen-14B, and DeepSeek-R1-Distill-Llama-8B.

**Questions:**

Please see weaknesses. I'd be happy to adjust my score after reading sufficient information about the dataset used for fine-tuning.

**Ethical Concerns:**

["NO or VERY MINOR ethics concerns only"]

**Final Justification:**

Authors addressed all my questions and concerns in the rebuttal.

Originally when I downloaded their supplementary material, the pdf file would not open, perhaps because the download on my end was corrupted. In the rebuttal, authors suggested I download the file again. This time I was able to see the supplementary materials and detailed information about the dataset used for fine-tuning. I am impressed to hear that authors have made sure there is no overlap between the training and test sets.

Overall, the method proposed in the paper is novel and useful, both for research and practice. The idea behind data generation is smart and fast for practical purposes. The empirical results reported by the paper is also impressive.

I suggested to authors to experiment with more LLMs. They promised to include results on Qwen3-30B-A3B-Thinking model. If such result is positive, I would give a score of 5.5 or 6 to the paper. Since that result is still unknown, for now, I give a score of 5.

**Limitations:**

The checklist indicates that limitations are discussed in the appendix, but I couldn't locate the appendix. Even in the supplementary materials, I did not see a pdf file.

**Paper Formatting Concerns:**

I didn't notice any issues.

**Quality:**

3

**Strengths And Weaknesses:**

### Strengths

The method itself is intuitive and well implemented. It can be useful for the community as it both reduces the computational cost and increases the accuracy. Table 1 is impressive.

Paper is well written. Motivation is clear and interesting.

Posing the problem of overthinking as a binary classification task and formulating it as such is a sound and interesting approach in my view.

&nbsp;&nbsp;


### Weaknesses:

The models used in the experiments are very similar. All of them are DeepDeek-R1-Distill models. And their size slightly varies. To convince a reader about the advantage of the proposed method, one would need to see experiments on other LLMs.

-------

Very little is explained about the self-constructed dataset used for training the model. Page one mentions that dataset is explained in the Appendix, but I can only see the checklist and no appendix at the end of the paper.

Are authors planning to release their fine-tuning dataset?

-------

Sensitivity to the verification task is not investigated. It is not clear if the dataset was a different dataset with a different verification task, the results would have been similar. Is there any overlap between the datasets used for testing and the fine-tuning dataset? How did the authors create their dataset? Without adequate information, it will be difficult to evaluate the usefulness of the method.

-------

In my view, the paper has considerable redundancy and repetitions. Starting from page 3, I felt that problem is already explained and I’m reading text that is mostly repetitive. At some points throughout the paper, the text asks questions from the reader. If you just isolate those questions, the repetitions become more clear.

---

> ### Author Rebuttal · Authors · 2025-07-30
>
> Thanks for the insightful and valuable feedback. We have provided detailed responses below and hope they address your concerns. If you have any further questions or require additional clarification, we would be glad to continue the discussion. We really appreciate your time and consideration.
>
> > **Q1: The models used in the experiments are very similar. All of them are DeepDeek-R1-Distill models.**
>
> **A:** Thanks for the valuable suggestions. To better demonstrate the effectiveness of our method on different LLMs, we also applied our proposed VeriThinker approach to Phi-4-mini-reasoning-3.8B [1] (produced by Microsoft), which has a different architecture and parameter size. As shown in Table 1, our method significantly reduces the token usage of Phi-4-mini-reasoning-3.8B while maintaining its original reasoning accuracy.
> #### *Table 1 Performance comparison on other LLM.*
> | Method| MATH500 | AIME24 |AIME25 |
> |--|--|--|--|
> | Phi-4-mini-reasoning-3.8B [1]| 92.8% (3554 tokens)| 46.3% (13108 tokens) | 34.6% (14261 tokens)|
> | **+VeriThinker (ours)** | **93.0% (2495 tokens)**| **45.9% (9204 tokens)** | **36.7% (9991 tokens)**|
>
>
> > **Q2: Very little is explained about the self-constructed dataset used for training the model.**
>
> **A:** Thanks for the thoughtful comments. We would like to clarify that all details of our dataset construction process are provided in **Section B of appendix.pdf** in the **supplementary.zip** submitted with our paper. We have confirmed that appendix.pdf is included in the supplementary.zip as downloaded from OpenReview. If you are unable to access it, this may be due to a technical issue on the platform, and we kindly suggest downloading the file again.
>
> Our data construction pipeline includes 4 steps:
> - **Step 1: Problem Collection.** We gather problems from the training sets of four mathematical datasets (PRM12K, GSM8K, LIMO, and Numina-Math) to ensure diversity in topics and difficulty. Specifically, we extract all problems from the training set of PRM12K, GSM8K, and LIMO. For Numina-Math, we only select problems whose solutions are integer-valued math-word-problems, simplifying correctness labeling.
> - **Step 2: CoT Solution Collection.** We generate multiple short-CoT solutions for each problem using five small language models (*Qwen-2.5-0.5B/1.5B/7B-Instruct*, *Qwen-Math-1.5B/7B-Instruct*). These solutions are step-by-step and do not include any self-reflection.
> - **Step 3: Correctness Labeling.** We label each solution as correct or incorrect by comparing its final answer to the ground truth, using the Hugging Face *math\_verify* tool for automated labeling.
> - **Step 4: Verification Data Filtering and Selection.** Initially, we discard problems where all five CoT solutions are uniformly correct or incorrect. Such problems lack informative training signals due to being either too trivial or excessively challenging, and this process also helps filter out inherently problematic data. Next, we apply a deduplication strategy using *Qwen-Math-1.5B-Instruct* as a reference model. Specifically, for problems correctly solved by the reference model, we retain its correct solutions along with incorrect solutions generated by the other models. Conversely, for problems incorrectly solved by the reference model, we retain its incorrect solutions and also incorrect solutions from the other models. This selection strategy ensures each problem contributes at least one correct and one incorrect CoT solution. This deduplication and selection process results in a fine-tuning dataset of 340K instances (150K correct and 190K incorrect solutions), formatted as shown in Appendix Figure 1
>
> This pipeline is highly efficient, the full dataset can be generated in 48 hours using just eight A5000 GPUs.
>
> We hope this clarifies our data construction process. Please refer to the appendix.pdf for full details.
>
>
> > **Q3: It is not clear if the dataset was a different dataset with a different verification task, the results would have been similar. Is there any overlap between the datasets used for testing and the fine-tuning dataset?**
>
> **A:** Thanks for the thoughtful comments. We confirm that our training and test sets are completely separated, with no risk of data leakage. To further demonstrate the robustness of VeriThinker, we conducted two additional experiments.
>
> First, we built a small CoT verification dataset using only PRM12K and GSM8K problems for training, and then evaluated the fine-tuned model on the much harder AIME24 and AIME25 datasets. As shown in Table 2, even with training data limited to PRM12K and GSM8K, our method effectively reduced token usage while maintaining or even improving accuracy on these more challenging benchmarks.
> #### *Table 2 Finetuned on PRM12K and GSM8K, evaluated on the much more difficult AIME Datasets*
> | Model| MATH500 | AIME24 |AIME25 |
> |--|--|--|--|
> | R1-Distill-7B | 94.0% (3791 tokens)| 54.1% (13108 tokens) | 38.7% (14321 tokens)|
> | **+VeriThinker**  | **95.0% (2534 tokens)**|**56.4% (10302 tokens)** | **39.8% (11270 tokens)**|
> | R1-Distill-14B| 95.2% (3529 tokens)| 69.0% (11724 tokens) | 49.6% (13409 tokens)|
> | **+VeriThinker**| **95.2% (2505 tokens)**| **68.7% (8425 tokens)** | **49.2% (10580 tokens)**|
>
> Secondly, to further illustrate this point, we evaluated our model (finetuned exclusively on mathematical problems) on the GPQA Diamond dataset. GPQA Diamond is a multiple-choice benchmark covering diverse disciplines (chemistry,physics, biology......) and significantly differs in both format and domain from our training data.  As shown in Table 3, our method achieved substantial CoT compression while maintaining accuracy on this challenging benchmark.
> #### *Table 3 Cross-Domain Performance on GPQA-Diamond*
> | Model| GPQA-Diamond |
> |--|--|
> | R1-distill-7B | 51.01% (7407 tokens)|
> | **+VeriThinker**  | **50.00% (4984 tokens)**|
> | R1-distill-14B | 60.1% (6996 tokens)|
> | **+VeriThinker**  | **60.6%  (4309 tokens)**|
>
> The above results demonstrate that our method still achieves strong performance even when there is a domain gap between the training and test sets. This cross-domain generalization is also a key strength of our approach.
>
> > **Q4: redundancy in paper:**
>
> **A:** Thanks for the valuable and insightful comments. We truly appreciate your feedback and will make sure to address this issue thoroughly in the next version of our paper. Your comments are very helpful in improving the quality of our work.
>
> ### Reference
> [1] "Phi-4-mini-reasoning: Exploring the limits of small reasoning language models in math." arXiv preprint arXiv:2504.21233 (2025).

---

> > ### Comment · Reviewer_c9kd · 2025-08-01
> >
> > I thank the authors for their clear and detailed response.
> >
> > I confirm that I can see the appendix and supplementary materials after downloading the file again. Thanks for the suggestion.
> >
> > About the new results on Phi-4-mini-reasoning-3.8B, I see that with VeriThinker, the accuracy decreases slightly for AIME24. However, with all the previous three LLMs, VeriThinker increased the accuracy while reducing the token budget. Is that correct? Do authors have any interpretation of why the accuracy decreases only for Phi-4-mini-reasoning-3.8B on AIME24? Could it be because Phi-4-mini-reasoning-3.8B is half the size of other three LLMs? Still, I see an accuracy gain by VeriThinker with Phi-4-mini-reasoning-3.8B on AIME25 and MATH500.
> >
> > After reading the authors' rebuttal and also the appendix, my concerns about most issues are now resolved, and I will increase my score.
> >
> > My only remaining feedback is: adding experiments on more LLMs of size at least 7B would make the results section stronger in my view. You have already experimented with three LLMs from DeepSeek. I'd suggest trying to add experiments on other LLMs for the final version of your paper.

---

> > > ### Author Response · Authors · 2025-08-01
> > > **Thanks for agreeing to increase the score and for your thoughtful feedback.**
> > >
> > > **Thanks for agreeing to increase the score and for your thoughtful feedback.**
> > >
> > > Regarding the slight performance differences between phi-4-mini-reasoning (3.8B) and R1-Distill-7/14B, we find that the gap primarily stems from their parameter scale. Specifically, we observed that the smaller 3.8B reasoning model, due to its limited parameter capacity, exhibits a slight risk of forgetting during fine-tuning on verification tasks, which marginally impacts its effectiveness. Nevertheless, models with 7b parameters and above are not subject to this risk.
> > >
> > > Additionally, we are also in the process of applying our method to Qwen3-30B-A3B-Thinking model, a much larger and MoE-based LLM. However, due to current computational resource constraints and time limitations, these results are not yet available. We will include them in the next version of our work.
> > >
> > > Once again, we truly appreciate your valuable suggestions and kindly encouragement, which have greatly benefited our research.

---

### Official Review · Reviewer_FWSD · 2025-07-01

**Clarity:** 3
**Significance:** 2
**Originality:** 3
**Rating:** 4
**Confidence:** 4

**Summary:**

The paper proposes a novel approach to enable concise reasoning, called VeriThinker. Instead of directly training on short CoTs, it trains the model to verify solutions effectively, which results in fewer solution changes and improved accuracy.

**Questions:**

Please refer to the weaknesses for details. Here are some key points:

- Any ablation on the number of verifier datasets?
- Any training details for the baselines?
- Any comparisons that match accuracy and compare compression rates instead?

**Ethical Concerns:**

["NO or VERY MINOR ethics concerns only"]

**Final Justification:**

As outlined in the review, the idea itself seems interesting. However, the evaluation setting is not clear. The authors provided a detailed response, but it did not fully justify their design choices. Therefore, I believe borderline acceptance is the right score.

**Limitations:**

yes

**Quality:**

3

**Strengths And Weaknesses:**

**Strengths:**

- The motivation is strong, although somewhat obvious, and the focus on verification is quite interesting.
- Sections 3.2 and 3.3 offer insightful analyses that justify the method well.
- The paper is easy to read, and I personally appreciated the narrative of how the authors arrived at their method.

**Weaknesses:**

The primary weakness, in my opinion, lies in the experimental setup, which lacks sufficient detail and raises concerns about reliability.

- **Data Construction for Training with Verifier Objectives:**
    - This is a crucial aspect, yet the main paper omits important details. Even in the appendix, the rationale for using five models and the number of generations is not clearly justified. I would also like to see how the size of the verification dataset affects performance.
    - While the authors note that synthetic data generation is costly, constructing a large verification dataset is also non-trivial. Although each CoT is short, unlike longer reflective reasoning, it’s still a significant effort as you generated a lot of data pairs.
    - Moreover, there’s no guarantee that each CoT step is correct, even if the final answer is.
- **Lack of Baseline Training Details:**
    - For baselines such as SFT, what data was used? Am I missing something? If different datasets were used for SFT and other baselines, can we consider the comparison fair? VeriThinker uses quite a variety of datasets, and such differences matter, especially considering prior work that maintains accuracy while significantly improving compression rates [1].
    - What does “comparable CoT compression for baselines” mean? Is there any way to control the level of compression in baselines? If so, corresponding graphs or analyses should be included.
- **Statistical Significance:**
    - Statistical significance is not reported, which is especially important for datasets like AIME 2024 and 2025.

**References**

[1] Munkhbat et al. *Self-Training Elicits Concise Reasoning in Large Language Models*

---

> ### Author Rebuttal · Authors · 2025-07-30
>
> Thanks for the thoughtful and valuable feedback. Please find our responses below, we hope they adequately address your concerns. If you have any further questions or need additional clarification, we’d be happy to discuss them with you. Thanks again for your time and efforts.
>
> > **Q1: the rationale for using five models and the number of generations is not clearly justified. I would also like to see how the size of the verification dataset affects performance.**
>
> **A:** Thanks for the insightful comments. We would like to clarify this issue.
>
> In our data construction procedure, we first generated CoT solutions for each problem using five different short-CoT models with greedy decoding (one solution per model per problem). Using multiple models of varying reasoning abilities allowed us to produce diverse reasoning paths for the same question, also ensuring almost every problem had both correct and incorrect solutions. Such diverse and contrastive data help the model robustly distinguish between correct and incorrect CoTs, thus benefiting efficient reasoning.
>
> To validate this, we conducted two additional experiments:
> * We trained the R1-Distill-7B model on a raw dataset (290k samples) generated using only one short-CoT model (Qwen-math-1.5b-instruct), instead of five.
> * We also trained the same model on a smaller dataset (32k samples) constructed solely from PRM12K and GSM8K using our original pipeline.
>
> As shown in Table 1:
> * The model trained on the single-model raw dataset performed worse in both token reduction and accuracy retention compared to our carefully constructed five-model dataset (340k samples), confirming the effectiveness of our approach.
> * The model trained on the smaller dataset achieved good accuracy but lower token reduction, highlighting the impact of dataset size on our method's performance.
>
> #### *Table 1 Performance comparison using different training data*
> | Method| MATH500 | AIME24 |AIME25 |
> |--|--|--|--|
> | R1-Distill-7B | 94.0% (3791 tokens)| 54.1% (13108 tokens) | 38.7% (14321 tokens)|
> | +VeriThinker w/ raw dataset (290k) | 94.8% (2349 tokens)| 52.7% (12023 tokens) | 36.7% (13286 tokens)|
> | +VeriThinker w/ small dataset (32k) | 95.0% (2534 tokens)| 56.4% (10402 tokens) | 39.8% (11270 tokens)|
> | +VeriThinker w/ our dataset (340k) | 94.8% (2125 tokens)| 56.5% (9381 tokens) | 40.8% (10287 tokens)|
>
>
> > **Q2: the cost of short-cot data pairs generation.**
>
> **A:** Thanks for the valuable comments. Generating large-scale datasets typically relies on inference engines like vLLM [3], which support batch inference. However, inference speed in batch scenarios heavily depends on the longest sequence within each batch. For instance, the Qwen-Math-7B-instruct model produces CoT solutions for the PRM12K dataset with a maximum token length of only 1K tokens, making it 20x faster than the R1-Distill-7B model, which has a maximum token length of 16K tokens. Additionally, short-CoT sequences consume significantly less memory, enabling much larger batch sizes and further accelerating data generation.
>
> To illustrate, using vLLM, our entire CoT verification dataset (initially 1.45M samples, 340K after filtering) was generated in just 48 hours on eight A5000 GPUs. These results clearly demonstrate the convenience and efficiency of our short-CoT data collection process.
>
>
> > **Q3: there’s no guarantee that each CoT step is correct, even if the final answer is.**
>
> **A:** Thanks for the insightful comments. All our generated solutions are short-CoT, meaning only have one single reasoning path without any self-reflection or branching. As a result, if a mistake occurs at any step, the model cannot recover and will almost always produce an incorrect final answer.
>
> To demonstrate this, we conducted an additional experiment: We randomly selected 50 correct short-CoT solutions from Qwen-Math-7B-instruct on PRM12K and introduced a calculation error at a random step in each chain. After using the Qwen-Math-7B-instruct model to continue reasoning from that step, every final answer was incorrect. This result clearly supports our point.
>
>
> > **Q4: For baselines such as SFT, what data was used?**
>
> **A:** Thanks for the valuable comments. For the SFT baseline, we followed the CoT-Valve [1] to synthesize concise CoT chains: we merged the base and reasoning models at a 0.2:0.8 ratio, then sampled concise CoT responses on the PRM12K and GSM8K datasets. We selected the correct CoT responses as SFT data. These concise CoTs are about 60% the length of the original chains. Both SFT and our VeriThinker method are finetuned using LoRA with same configuration.
>
> To ensure a fair comparison, we also constructed a small verification dataset using only PRM12K and GSM8K, and trained our SVFT method on this data. As shown in Table 2, our method consistently outperforms the SFT baseline, especially on the more challenging AIME datasets, where SFT methods always suffer from a significant drop in accuracy.
>
> Importantly, generating concise CoT data for SFT is much more expensive than collecting verification data for our method
> #### *Table 2 Performance comparison with SFT on the same training dataset.*
> | Method| MATH500 | AIME24 |AIME25 |
> |--|--|--|--|
> | R1-Distill-7B| 94.0% (3791 tokens)| 54.1% (13108 tokens) | 38.7% (14321 tokens)|
> | +SFT (PRM12K+GSM8K)| 91.2% (2064 tokens)| 49.1% (8843 tokens) | 32.1% (9686 tokens)|
> | **+VeriThinker(PRM12K+GSM8K)**| **95.0% (2534 tokens)**| **56.4% (10402 tokens)** | **39.8% (11270 tokens)**|
> | R1-Distill-14B| 95.2% (3529 tokens)| 69.0% (11724 tokens) | 49.6% (13409 tokens)|
> | +SFT (PRM12K+GSM8K) | 92.8% (2226 tokens)| 58.3% (9021 tokens) | 43.3% (10644 tokens)|
> | **+VeriThinker(PRM12K+GSM8K)**| **95.2% (2505 tokens)**| **68.7% (8425 tokens)** | **49.2% (10580 tokens)**|
>
>
> > **Q5: What does “comparable CoT compression for baselines” mean? Is there any way to control the level of compression in baselines?**
>
> **A:** Thanks for the comments. The compression rate of our VeriThinker cannot be controlled. However, the compression rate of the baseline SFT-based method can be controlled. For SFT, we control the compression rate by adjusting the length of the synthetic concise chains used as training data. The synthetic concise chains we used for SFT are approximately 60% of the original length, which closely matches our VeriThinker's compression rate. This allows us to compare both methods at comparable compression levels, ensuring a fair and meaningful accuracy evaluation.
>
> > **Q6: Statistical significance is not reported, which is especially important for datasets like AIME 2024 and 2025.**
>
> **A:** Thanks for the valuable suggestions. For the AIME dataset, the accuracy we report is calculated by running inference on each question 16 times and then taking the average. In Table 3 below, we present both the mean accuracy and the standard deviation to demonstrate the statistical significance of our results.
> #### *Table 3 Statistical significance on AIME datasets*
> |Method|AIME24|AIME25|
> |-|-|-|
> | R1-Distill-7B |  54.1%+-6.2% (13108 tokens) | 38.7%+-3.3% (14321 tokens)|
> | +VeriThinker  |  56.5%+-7.0% (9381 tokens) | 40.8%+-5.9% (9686 tokens)|
> | R1-Distill-8B |  44.6%+-5.2% (14005 tokens) | 30.1%+-4.6% (14420 tokens)|
> | +VeriThinker  |  46.9%+-5.7% (11285 tokens) | 29.7%+-6.3% (10557 tokens)|
> | R1-Distill-14B |  69.0%+-6.8% (11724 tokens) | 49.6%+-5.2% (13409 tokens)|
> | +VeriThinker  |  73.0%+-6.0% (7423 tokens) | 54.8%+-5.5% (9304 tokens)|
>
>
> > **Q7: Any comparisons that match accuracy and compare compression rates instead?**
>
> **A:** Thanks for the comments. Since SFT-based CoT compression methods inevitably cause a significant drop in accuracy on challenging tasks (AIME) and our method can maintain the accuracy, we conduct an additional comparison with the advanced RL-based method AdaptThink[2] to better match accuracy and compare compression rates. As shown in Table 4, when both methods maintain high accuracy on AIME, our approach achieves lower token usage compared to AdaptThink.
>
> #### *Table 4 Comparing compression rates under similar accuracy*
> | Model| MATH500 | AIME24 |AIME25 |
> |--|--|--|--|
> | R1-Distill-7B | 94.0% (3791 tokens)| 54.1% (13108 tokens) | 38.7% (14321 tokens)|
> | +AdaptThink [2] | 91.6% (2080 tokens)| 53.3% (9998 tokens) | 38.75% (11330 tokens)|
> | **+VeriThinker (ours)** | **94.8% (2125 tokens)**| **56.5% (9381 tokens)** | **40.8% (10287 tokens)**|
>
> ### Reference
> [1] CoT-Valve: Length-Compressible Chain-of-Thought Tuning. In Proceedings of the 63rd Annual Meeting of the Association for Computational Linguistics.
>
> [2] "Adaptthink: Reasoning models can learn when to think." arXiv preprint arXiv:2505.13417 (2025).
>
> [3] "Efficient memory management for large language model serving with pagedattention." Proceedings of the 29th symposium on operating systems principles. 2023.

---

> ### Comment · Area_Chair_Kks4 · 2025-08-05
>
> The authors have provided a rebuttal to your comments, and it's an important part of the review process to give their response careful consideration. Please take a moment to review their rebuttal and provide any follow-up comments. This will help ensure there’s sufficient time for discussion and any necessary follow-up.
>
> Best regards,
>
> AC

---

> ### Comment · Reviewer_FWSD · 2025-08-06
> **Response to the Authors**
>
> Thanks for the detailed rebuttal, and apologies for the delayed response due to some eye issues.
>
> Regarding Q1, I believe the paper would be stronger if it were less sensitive to the dataset. If certain methods require a large amount of diverse reasoning data, is that really a desirable property?
>
> For Q2-Q5, I’m still not fully convinced. The main issue is whether the comparison is fair for SFT. What if we also used different models for SFT, not just DeepSeek-R1, which produces long CoTs, and then applied RFT? Wouldn’t that also lead to lower generation costs? In Table 4, it seems that for the 7B model, SFT reduces token usage significantly in additional experiments.
>
> In short, my concern is about fair and controlled settings. I think the paper would be clearer if you ran Verithinker in a minimal setting: generate synthetic data using the same model on the same dataset for both SFT and Verithinker, then plot a graph with the x-axis as token length and y-axis as accuracy for SFT, and show Verithinker as a point on the same graph (like a Pareto frontier). Right now, the mixture of different datasets and settings makes it harder to clearly interpret the effectiveness of the method.
>
> That said, I understand this may not be something that can be fully addressed in the rebuttal phase. I will likely maintain my score. Still, I consider this a borderline accept, as the focus on verification is quite interesting.

---

> > ### Author Response · Authors · 2025-08-07
> >
> > Thanks for carefully reviewing our rebuttal and for your valuable feedback. We greatly appreciate your professionalism and dedication throughout the review process. We would like to take this opportunity to provide further clarification.
> >
> > > **Q1. The sensitive to the dataset**
> >
> > Thanks for the insightful comments. You are correct that our dataset construction indeed requires multiple models to generate many solutions. However, we wish to clarify that this requirement does not lead to inconvenience in practice. For reference, constructing our entire dataset only took 48 hours on 8 A5000 GPUs, it is a one-time investment, after which the dataset can be leveraged for all downstream reasoning models. Utilizing this dataset, our method has demonstrated strong performance across multiple models, including R1-Distill-Qwen-7B, R1-Distill-Qwen-14B, and R1-Distill-Llama-8B. Moreover, as shown in Table 1, this dataset also achieves excellent results with the Phi-4-mini-reasoning-3.8B model [1] which has a different architecture. Thus, users can directly fine-tune their own reasoning LLMs using our dataset, without needing to spend additional effort on dataset construction. In comparison, many existing methods based on SFT or RL typically need to generate separate synthetic datasets tailored for each specific model.
> > #### *Table 1 Performance comparison on other LLM.*
> > | Method| MATH500 | AIME24 |AIME25 |
> > |--|--|--|--|
> > | Phi-4-mini-reasoning-3.8B [1]| 92.8% (3554 tokens)| 46.3% (13108 tokens) | 34.6% (14261 tokens)|
> > | **+VeriThinker (ours)** | **93.0% (2495 tokens)**| **45.9% (9204 tokens)** | **36.7% (9991 tokens)**|
> >
> > Additionally, our method exhibits strong generalizability across dataset domains. As illustrated in Table 2, although our model was trained exclusively on self-constructed mathematical verification datasets, it performs very well on the GPQA Diamond dataset. GPQA Diamond is a multiple-choice benchmark that encompasses diverse fields such as chemistry, physics, and biology, and it substantially differs from our training data in both format and domain. This outcome indicates that our method is not sensitive to domain differences between training and test sets.
> > #### *Table 2 Cross-Domain Performance on GPQA-Diamond*
> > | Model| GPQA-Diamond |
> > |--|--|
> > | R1-distill-7B | 51.01% (7407 tokens)|
> > | **+VeriThinker**  | **50.00% (4984 tokens)**|
> > | R1-distill-14B | 60.1% (6996 tokens)|
> > | **+VeriThinker**  | **60.6%  (4309 tokens)**|
> >
> > [1] Xu, Haoran, et al. "Phi-4-mini-reasoning: Exploring the limits of small reasoning language models in math." arXiv preprint arXiv:2504.21233 (2025).
> >
> > > **Q2. it seems that for the 7B model, SFT reduces token usage significantly in additional experiments.**
> >
> > Thanks for highlighting this important point. You are correct that SFT significantly reduces token usage; however, this reduction comes at the cost of notably decreased accuracy. Our analysis shows that SFT indiscriminately lowers the model’s probability to reflect, irrespective of whether reasoning steps are correct or incorrect.
> >
> > We demonstrate this issue by analyzing the fine-tuned model's reflection behavior on the MATH500 dataset. Specifically, we measured the probability of generating a "reflection" token (e.g., "Wait") after correct and incorrect reasoning steps.
> >
> > As detailed in Table 3, both SFT and our method reduced the reflection probability for correct steps by approximately threefold. However, for incorrect steps, our method slightly increased the probability of reflection, whereas SFT continued to significantly reduce it. This indicates that SFT does not enable the model to discern necessary reflections from redundant ones; instead, it simply instills a general bias towards less reflection. In contrast, our method allows the model to more accurately determine when reflection is necessary.
> >
> > Consequently, on challenging datasets like AIME, our method achieves substantial reductions in token usage while preserving accuracy, whereas SFT experiences a significant drop in accuracy. Therefore, while SFT achieves token reduction, its benefits should be weighed against the significant trade-off in accuracy.
> >
> > #### *Table 3 Reflection probability when facing correct and incorrect steps*
> > | Model| Correct Steps | Incorrect Steps |
> > |--|--|--|
> > | R1-Distill-7B | 11.1%| 36.6%  |
> > | +SFT  | 3.8% | 18.8%  |
> > | **+VeriThinker**  | **4.1%** | **37.6%**  |

---

> > ### Author Response · Authors · 2025-08-07
> >
> > > **Q3. I think the paper would be clearer if you ran Verithinker in a minimal setting: generate synthetic data using the same model on the same dataset for both SFT and Verithinker**
> >
> > Thanks for the valuable suggestion. In accordance with your suggestion, we conducted a completely fair comparative experiment. Specifically, both our VeriThinker and the SFT method used synthetic data generated from the same low-cost short-CoT models (qwen-2.5-0.5b/1.5b/7b-instruct, qwen-math-1.5b/7b-instruct) on the same datasets (PRM12K, GSM8K, LIMO, and NuminaMath). We also ensured that all training configurations were kept exactly the same for both methods.
> >
> > As shown in Table 4, under these strictly controlled settings, although SFT substantially reduced token usage, it suffered a dramatic decline in accuracy, failing to solve most of the problems on the AIME dataset. This occurs because solutions generated by short-CoT models lack self-reflection steps. Training with such data causes the reasoning LLM to **degrade** into a non-reasoning model incapable of self-reflection, thus losing the accuracy improvements typically gained through test-time scaling. Consequently, SFT data needs to be synthesized using much more expensive long-CoT models, which significantly increases the cost of data generation.
> >
> > In contrast, our proposed VeriThinker approach reduces CoT length significantly while maintaining or even slightly enhancing accuracy, even using only cheap short-CoT data. This fair comparison demonstrates the advantages of our proposed method and explains why SFT data generation requires costly long-CoT models.
> >
> > #### *Table 4 Performance comparison with SFT*
> > | Method| MATH500 | AIME24 |AIME25 |
> > |--|--|--|--|
> > | R1-Distill-7B | 94.0% (3791 tokens)| 54.1% (13108 tokens) | 38.7% (14321 tokens)|
> > | +SFT  | 81.4% (780 tokens)| 13.7% (2729 tokens) | 8.3% (2133 tokens)|
> > | **+VeriThinker**  | **94.8% (2125 tokens)**| **56.5% (9381 tokens)** | **40.8% (10287 tokens)**|
> >
> > Once again, we sincerely appreciate your insightful comments and your recognition of our work. Your feedback has greatly helped us to strengthen our work. We hope that these additional analyses will address your concerns and further strengthen your confidence in our work.

---

### Official Review · Reviewer_ZRxU · 2025-07-01

**Clarity:** 2
**Significance:** 2
**Originality:** 2
**Rating:** 4
**Confidence:** 5

**Summary:**

In this paper, the authors propose VeriThinker to address the issue of overthinking in LLMs. The core idea of VeriThinker is to train LLMs to predict the correctness of CoTs. The authors argue that equipping LLMs with this capability enables them to avoid unnecessarily lengthy responses when the solution is correct, while still retaining the ability to revise incorrect ones. They also develop a solution-wise speculative reasoning mechanism based on this insight. Experiments on three benchmarks demonstrate that VeriThinker maintains—or even slightly improves—reasoning accuracy while reducing the number of output tokens.

**Questions:**

1. In line 138, the authors state that "The prevalence of overthinking stems from insufficient accuracy in this binary classification task." However, there is no evidence or existing work provided to support this statement. I am very skeptical of this claim, as many reasoning models are well capable of judging the correctness of CoTs.

2. In line 147, it is unclear why "these methods synthesize a concise target chain approximating the optimal scenario where p(acc | h) ≈ 100%." First, I do not understand why the authors emphasize that "current models that employ synthetic target chains for CoT implicitly aim to maximize p(acc | h)." Don’t other models also aim to maximize accuracy? Second, there is no justification for why a concise target chain leads to the optimal scenario.

3. In Section 3.3, the authors find that, apart from Dataset (5), all other datasets result in maintained or slightly improved accuracy. However, this result is strange because, in Dataset (2), the labels for solution correctness are completely reversed, i.e., correct solutions are labeled as "wrong". This phenomenon suggests that the model’s ability to predict correctness does not matter at all, which contradicts the main hypothesis of the paper.

4. The authors need to conduct an ablation study on the training data, as the current experimental setup raises concerns about whether the accuracy improvement comes from the model having seen the correct answers during training (even if not trained on them directly). For example, the Numina-Math dataset includes AIME data. It is also important to test whether the model can classify the correctness of solutions to problems it has never encountered before (e.g., more difficult problems), which is crucial for evaluating the generalization ability of the proposed method.

5. In section 4.2 "CoT Correctness Verification Results", the authors need to add more baselines to compare how well a basic LLM performs on the correctness verification task.

**Ethical Concerns:**

["NO or VERY MINOR ethics concerns only"]

**Final Justification:**

As the authors provide sufficient experiments in the rebuttal phase, which address my concerns (especially the concern about self-verification), I think this paper poses great value to the community. Therefore, I have increased the score to Borderline accept.

**Limitations:**

Yes

**Paper Formatting Concerns:**

No.

**Quality:**

3

**Strengths And Weaknesses:**

Strengths:
1. The paper is clearly written and easy to understand.
2. The proposed method shows potential in balancing reasoning accuracy with computational cost.

However, several important issues remain:
1. Many key claims lack sufficient evidence or support, which weakens the overall motivation and persuasiveness of the paper (please refer to Q1 and Q2 below).
2. Many experimental results are questionable and require further clarification or validation (please refer to Q3-Q5 below).
3. The experiments do not include comparisons with recent RL-based methods, which limits the completeness of the evaluation.
4. There are a number of typos. For example:
– Line 61: "represent" should be "represents".
5. The paper is overly verbose in several sections, leading to redundancy and reduced clarity. For instance, in Section 4.2 "CoT Compression Results", the conclusions in two consecutive paragraphs are essentially identical, which gives the impression of filler content.

---

> ### Author Rebuttal · Authors · 2025-07-30
>
> Thanks for the insightful and valuable feedback. We have provided detailed responses below and hope they address your concerns. If you have any further questions or require additional clarification, we would be glad to continue the discussion. We really appreciate your time and consideration.
>
> > **Q1: The statement of "The prevalence of overthinking stems from insufficient accuracy in this binary classification task." is not supported. I am very skeptical of this claim, as many reasoning models are well capable of judging the correctness of CoTs.**
>
> **A:** Thanks for the insightful comments. Prior work [1,2,3] has also found that reasoning models often struggle to judge the correctness of their own CoT outputs, frequently triggering unnecessary self-reflection even when previous steps are correct. We further investigated this issue by analyzing R1-Distill-7B’s behavior. Specifically, we measured the probability of generating a "reflection" token (such as "Wait") after correct and incorrect reasoning steps on the MATH500 dataset.
>
> As shown in Table 1, even after fully correct steps, large reasoning models initiate self-reflection more than 11% of the time. For incorrect steps, this probability jumps to 36.8%. After fine-tuning with our method, the reflection rate on correct steps drops by nearly threefold, while the rate for incorrect steps increases slightly. These results highlight the core problem and demonstrate how our approach effectively reduces unnecessary self-reflections without sacrificing reasoning accuracy.
> #### *Table 1 Reflection probability when facing correct and incorrect steps*
> | Model| Correct Steps | Incorrect Steps |
> |--|--|--|
> | R1-Distill-7B | 11.1%| 36.6%  |
> | +VeriThinker  | 4.1% | 37.6%  |
>
>
>
> > **Q2: It is unclear why "these methods synthesize a concise target chain approximating the optimal scenario where p(acc | h) ≈ 100%."**
>
> **A:** Thanks for the comments. We apologize for the confusion and would like to clarify this point.
>
> In our paper, $p(\text{acc}|h)$ does not refer to the probability that the generated CoT is correct. Instead, it represents the probability that the model correctly identifies whether reflection is necessary at each reasoning step. Ideally, models should avoid reflecting on correct steps while always reflecting when encountering incorrect ones, corresponding to $ p(\text{acc}|h) = 100\%$.
>
> Previous CoT compression methods indirectly improved reflection accuracy by training models to imitate concise chains, which contain minimal, necessary reflections. In contrast, our method directly fine-tunes the model to explicitly distinguish correct from incorrect reasoning steps. This direct training significantly enhances the model’s judgment of when to reflect, as clearly supported by our experimental results in Table 1.
>
>
> > **Q3: In Section 3.3, the authors find that, apart from Dataset (5), all other datasets result in maintained or slightly improved accuracy. This result is strange.**
>
> **A:** Thanks for the comments. We appreciate the chance to clarify this further.
>
> In our SVFT method, the model is trained to discriminate between correct and incorrect CoT solutions. Importantly, the model does not know which group represents true correctness; it essentially learns to separate the two categories based on their features. The labels during training serve only as group identifiers and their inherent meaning does not matter. This is why, in Dataset (2), even if the labels are reversed, our method still works. In fact, as shown in paper section 3.3, our method performs just as well if we label the groups as "north" and "south" instead.
>
> After fine-tuning, the model's parameters are adjusted to become more sensitive to tokens that differentiate correct and incorrect CoTs. As a result, when deciding whether to self-reflect during reasoning, the finetuned model can more accurately identify when reflection is truly needed. Consequently, the model can avoid unnecessary self-reflections while keeping the essential ones. Our experimental results in Table 1 also confirm this point.
>
>
>
> > **Q4: The authors need to conduct an ablation study on the training data, as the current experimental setup raises concerns about whether the accuracy improvement comes from the model having seen the correct answers during training (even if not trained on them directly).**
>
> **A:** Thanks for the valuable comments. We conducted two additional experiments to clarify this issue.
>
> First, we constructed a small CoT verification dataset using only questions from PRM12K and GSM8K, and fine-tuned our reasoning model on this restricted data. We then evaluated the fine-tuned model on the more challenging AIME datasets. As shown in Table 2, even with training data limited to PRM12K and GSM8K, our method significantly reduced token usage and maintained accuracy on the AIME benchmarks.
> #### *Table 2 Finetuned on PRM12K and GSM8K, evaluated on the much more difficult AIME Datasets*
> | Model| MATH500 | AIME24 |AIME25 |
> |--|--|--|--|
> | R1-Distill-7B | 94.0% (3791 tokens)| 54.1% (13108 tokens) | 38.7% (14321 tokens)|
> | **+VeriThinker**  | **95.0% (2534 tokens)**| **56.4% (10302 tokens)** | **39.8% (11270 tokens)**|
> | R1-Distill-14B| 95.2% (3529 tokens)| 69.0% (11724 tokens) | 49.6% (13409 tokens)|
> | **+VeriThinker**| **95.2% (2505 tokens)**| **68.7% (8425 tokens)** | **49.2% (10580 tokens)**|
>
> Secondly, to further illustrate this point, we evaluated our model (trained exclusively on mathematical problems) on the GPQA Diamond dataset. GPQA Diamond is a multiple-choice benchmark covering diverse disciplines (chemistry,physics, biology......) and significantly differs in both format and domain from our training data. As shown in Table 3, our method still significantly compressed the length of CoTs while maintaining accuracy.
> #### *Table 3 Cross-Domain Performance on GPQA-Diamond*
> | Model| GPQA-Diamond |
> |--|--|
> | R1-Distill-7B | 51.01% (7407 tokens)|
> | **+VeriThinker**  | **50.00% (4984 tokens)**|
> | R1-Distill-14B | 60.1% (6996 tokens)|
> | **+VeriThinker**  | **60.6%  (4309 tokens)**|
>
> The above results demonstrate that our method still achieves strong performance even when there is a domain gap between the training and test sets.
>
> > **Q5: the authors need to add more baselines to compare how well a basic LLM performs on the correctness verification task.**
>
> **A:** Thanks for the comments. To further validate our approach, we used Qwen-2.5-1.5B-Instruct to generate CoT solutions on MATH500 and evaluated the verification accuracy of several models.
>
> We compared our VeriThinker-7B with two strong open-source models (Qwen-2.5-7B-Instruct and Llama-3.1-8B-Instruct) and two advanced closed-source models (OpenAI GPT-4o and GPT-4.1). As shown in Table 4, our method achieves substantially higher verification accuracy than the open-source models, and even outperforms the two large closed-source models from OpenAI.
> #### *Table 4 Comparison of CoT correctness verification capability*
> |Model|accuracy|pre|rec|F1|
> |-|-|-|-|-|
> | Qwen-2.5-7B-Instruct  | 74.8%| 0.714| 0.934| 0.810|
> | Llama-3.1-8B-Instruct | 70.0%| 0.668| 0.948 | 0.783|
> | OpenAI-GPT-4o  | 85.0% | 0.889| 0.843| 0.866|
> | OpenAI-GPT-4.1 | 86.4% | 0.910| 0.847 | 0.877|
> | **VeriThinker-7B (Ours)**| **90.4%**| **0.972**| **0.857**| **0.911**|
>
> > **Q6: The experiments do not include comparisons with recent RL-based methods**
>
> **A:** Thanks for the valuable comments. As shown in Table 5, we compare VeriThinker with two recent RL-based methods (AdaptThink [4] and LC-R1 [5]), both of which use improved GRPO algorithms for CoT compression. VeriThinker outperforms AdaptThink in both token reduction and accuracy. While LC-R1 achieves a higher compression rate, VeriThinker delivers much better accuracy.
>
> Importantly, RL-based methods require much higher computational resources due to the use of rollout strategies and full-model finetuning. For example, AdaptThink needs 32 H100 GPUs (80GB) for 28 hours of training, while VeriThinker achieves strong results with only 8 A5000 GPUs (24GB) and 14 hours of training.
> #### *Table 5 Comparison to recent RL-based methods*
> | Model| MATH500 | AIME24 |AIME25 |
> |--|--|--|--|
> | R1-Distill-7B | 94.0% (3791 tokens)| 54.1% (13108 tokens) | 38.7% (14321 tokens)|
> | +LC-R1 [5] | 89.6% (1425 tokens)| 47.5% (6495 tokens) | 35.4% (7523 tokens)|
> | +AdaptThink [4] | 91.6% (2080 tokens)| 53.3% (9998 tokens) | 38.75% (11330 tokens)|
> | **+VeriThinker (ours)** |**94.8% (2125 tokens)**| **56.5% (9381 tokens)** | **40.8% (10287 tokens)**|
>
> > **Q7: Typos and redundancy in paper:**
>
> **A:** Thanks for the thorough and thoughtful review. We are committed to addressing all of these issues comprehensively in the next version. We truly appreciate your valuable suggestions, which will certainly help us improve the quality of our paper.
>
> ### Reference
> [1] Can Large Language Models Detect Errors in Long Chain-of-Thought Reasoning?. In Proceedings of the 63rd Annual Meeting of the Association for Computational Linguistics (Volume 1: Long Papers), pages 18468–18489
>
> [2]"No free labels: Limitations of llm-as-a-judge without human grounding." arXiv preprint arXiv:2503.05061 (2025).
>
> [3] "Do not think that much for 2+ 3=? on the overthinking of o1-like llms." arXiv preprint arXiv:2412.21187 (2024).
>
> [4] "Adaptthink: Reasoning models can learn when to think." arXiv preprint arXiv:2505.13417 (2025).
>
> [5] "Optimizing Length Compression in Large Reasoning Models." arXiv preprint arXiv:2506.14755 (2025).

---

> > ### Comment · Reviewer_ZRxU · 2025-08-01
> >
> > Thank you for the detailed responses, especially on Q2 and Q3. They clarified most of my concerns.
> >
> > Regarding Q1, some prior works[1,2] also successfully use LLMs as verifiers to improve reasoning.
> >
> > This seems somewhat contradictory. Could you please conduct direct experiments comparing verifier accuracy and reflection rates to clarify this point?
> >
> > [1] Large Language Models are Better Reasoners with Self-Verification
> >
> > [2] SelfCheck: Using LLMs to Zero-Shot Check Their Own Step-by-Step Reasoning

---

> > > ### Author Response · Authors · 2025-08-02
> > >
> > > Thanks for acknowledging our rebuttal and for your valuable feedback. We would like to take this opportunity to offer further clarification.
> > >
> > > You are correct that prior works [1,2] have successfully used LLMs as self-verifiers, however, we wish to clarify that our viewpoint does not contradict these works. The key distinction lies in the type of models being studied. The two papers you cited focus on general-purpose LLMs (GPT-3, GPT-4) that do not incorporate a self-reflection step. In contrast, our research centers on recent, specialized reasoning models (deepseek-r1-distill), which are characterized by their self-reflection mechanisms and test-time scaling laws. A critical difference we observed is that, unlike general-purpose LLMs, these reasoning LLMs have very limited capability to directly verify the correctness of a given CoT solution. Instead of assessing the given CoT, the reasoning LLM tends to re-solve the problem from scratch with a significant number of tokens.
> > >
> > > Following your suggestion, we have also provided a presentation of verifier accuracy and reflection rates with additional experiments: For verifier accuracy evaluation, we generated CoT solutions on the MATH500 dataset using qwen-2.5-instruct-1.5B. We then used prompts to force the reasoning LLMs to directly judge the correctness of these solutions. For reflection rate evaluation, we used R1-Distill-7B to generate CoT solutions on MATH500. We then measured the reflection rate as the probability of the reasoning LLMs outputting a reflection token *after* the first correct solution in the generated CoT solutions.
> > >
> > > The results are shown in the table below. Compared to the original R1-Distill-7B, our proposed method leads to a significant improvement in the verifier accuracy while substantially reducing the reflection rate on already correct solutions.
> > >
> > > | Model| Verifier Accuracy ↑ | Reflection Rates ↓|
> > > |--|--|--|
> > > | R1-Distill-7B | 66%| 68%  |
> > > | **+VeriThinker**  | **90%**| **48%**  |
> > >
> > > We hope this addresses your concerns, and we are very grateful for your insightful comments and encouragement.
> > >
> > > ### Reference
> > >
> > > [1] Large Language Models are Better Reasoners with Self-Verification
> > >
> > > [2] SelfCheck: Using LLMs to Zero-Shot Check Their Own Step-by-Step Reasoning

---

> > > > ### Comment · Reviewer_ZRxU · 2025-08-03
> > > >
> > > > Thanks for you clarification. I am satisfied with the response, and will increase the scores.

---

> > > > > ### Author Response · Authors · 2025-08-03
> > > > > **Thanks for kindly agreeing to raise our score and for your thoughtful comments.**
> > > > >
> > > > > Thanks for kindly agreeing to raise our score and for your thoughtful comments. We sincerely appreciate the time and care you have devoted to reviewing our work. Your detailed comments are extremely valuable in helping us improve our paper. We will carefully incorporate your suggestions in the next revision.

---

### Decision · Program_Chairs · 2025-09-17

**Decision:**

Accept (poster)

**Comment:**

Reviewers agree the paper is clearly written, the method is intuitive and potentially useful, and results show reduced computation with maintained or slightly improved accuracy. The main concerns are about experimental completeness, dataset construction details, and limited model diversity. The rebuttal clarified several of these points, alleviating major doubts. Overall, I consider the paper technically solid and recommend acceptance.